# Conservative Prediction via Data-Driven Confidence Minimization

**Caroline Choi**[*]
*Department of Computer Science*
*Stanford University*

*cchoi1@cs.stanford.edu*

**Fahim Tajwar**[*]
*Machine Learning Department*
*Carnegie Mellon University*

*ftajwar@cs.stanford.edu*

**Yoonho Lee**[*]
*Department of Computer Science*
*Stanford University*

*yoonho@cs.stanford.edu*

**Huaxiu Yao**
*Department of Computer Science*
*University of North Carolina at Chapel Hill*

*huaxiu@cs.unc.edu*

**Ananya Kumar**
*Department of Computer Science*
*Stanford University*

*ananya@cs.stanford.edu*

**Chelsea Finn**
*Department of Computer Science*
*Stanford University*

*cbfinn@cs.stanford.edu*

**Reviewed on OpenReview:** *https://openreview.net/forum?id=QPuxjsjKCP*

## Abstract

In safety-critical applications of machine learning, it is often desirable for a model to be *conservative*, abstaining from making predictions on "unknown" inputs which are not well-represented in the training data. However, detecting unknown examples is challenging, as it is impossible to anticipate all potential inputs at test time. To address this, prior work (Hendrycks et al., 2018) minimizes model confidence on an auxiliary outlier dataset carefully curated to be disjoint from the training distribution. We theoretically analyze the choice of auxiliary dataset for confidence minimization, revealing two actionable insights: (1) if the auxiliary set contains unknown examples similar to those seen at test time, confidence minimization leads to provable detection of unknown test examples, and (2) if the first condition is satisfied, it is unnecessary to filter out known examples for out-of-distribution (OOD) detection. Motivated by these guidelines, we propose the Data-Driven Confidence Minimization (DCM) framework, which minimizes confidence on an *uncertainty dataset*. We apply DCM to two problem settings in which conservative prediction is paramount – selective classification and OOD detection – and provide a realistic way to gather uncertainty data for each setting. In our experiments, DCM consistently outperforms existing selective classification approaches on 4 datasets when tested on unseen distributions and outperforms state-of-the-art OOD detection methods on 12 ID-OOD dataset pairs, reducing FPR (at TPR 95%) by 6.3% and 58.1% on CIFAR-10 and CIFAR-100 compared to Outlier Exposure.

---

[*]Equal contribution.

# 1 Introduction

While deep neural networks have demonstrated remarkable performance on many tasks, they often fail unexpectedly (Simonyan & Zisserman, 2014; Zhang et al., 2017). In safety-critical domains such as healthcare, such errors may prevent the deployment of machine learning altogether. For example, a tumor detection model that is trained on histopathological images from one hospital may perform poorly when deployed in other hospitals due to differences in data collection methods or patient population (Koh et al., 2021). In these scenarios, it may be preferable to defer to a human expert. *Conservative* models—models that can refrain from making predictions when they are likely to make an error—may offer a solution.

Two fields of research aim to produce conservative models: selective classification (Liu et al., 2019; Kamath et al., 2020; Huang et al., 2020) and out-of-distribution (OOD) detection (Hendrycks & Gimpel, 2016; Liang et al., 2017a; Lee et al., 2018; Liu et al., 2020). In both problem settings, inputs can be seen as belonging to one of two high-level categories: *known* and *unknown* examples. Known examples are inputs that are

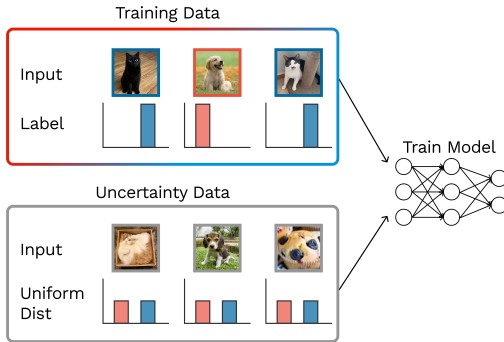

Figure 1: Data-driven confidence minimization (DCM) is a framework for training a model to make conservative predictions. DCM incorporates a regularizer that minimizes confidence on an unlabeled mixture of known and unknown examples that are similar to those seen at test-time.

well-represented in the training distribution. Unknown examples include inputs belonging to a new class not seen during training (OOD detection), and misclassified inputs insufficiently covered by the training distribution (selective classification). Despite considerable research in these areas, the problem of learning a conservative model remains challenging. As the test distribution can vary in a myriad of ways, it is impractical to anticipate the exact examples that will arise at test time.

A promising approach to OOD detection is Outlier Exposure (Hendrycks et al., 2018), which fine-tunes a pretrained model with a combined objective of standard cross-entropy on training examples and a regularizer that minimizes confidence on an auxiliary dataset, carefully curated to be distinct from the training distribution. Unlabeled auxiliary data is often readily available, making it an effective way of exposing the model to the types of unknown inputs it may encounter at test time.

Contrary to Outlier Exposure, which minimizes confidence on a specific choice of auxiliary dataset, we aim to better understand a broader class of approaches that minimize confidence on an *uncertainty dataset* and are applicable to both selective classification and OOD detection. Our theoretical analysis reveals two key guidelines for creating an effective uncertainty dataset. First, it should contain unknown examples that the model is likely to encounter and misclassify at test-time. Second, in the setting of OOD detection, if the first criteria holds, then the uncertainty set can also contain known examples, eliminating the need to filter out known examples as required by Hendrycks et al. (2018). In other words, it can be harmless to minimize confidence on known examples, which the model should be confident about. Our theory explains this counter-intuitive phenomenon: the effect of confidence minimization on known examples in the uncertainty set is "cancelled out" by the cross entropy loss on training examples, while the confidence loss on unknown examples in the uncertainty set is not. We show that using such an uncertainty set provably detects unknown inputs under mild assumptions.

Building on these insights, we present **Data-Driven Confidence Minimization (DCM)** as a unified approach for conservative prediction in selective classification and OOD detection. For each problem setting, we propose a realistic and effective way to construct the uncertainty dataset that follows the two guidelines above. For selective classification, we use misclassified examples from a held-out validation set from the training distribution as the uncertainty dataset, which naturally reflects what the model is likely to misclassify at test time. For OOD detection, we use an uncertainty dataset consisting of unlabeled examples from the test distribution, which includes both known and unknown examples. While it's not always the case that unlabeled examples from the test distribution are available, there are a number of real world applications

where this is the case, such as unannotated medical data from a new hospital (Sagawa et al., 2021). We visualize the DCM framework in Figure 1.

We empirically verify our approach through experiments on several standard benchmarks for selective classification and OOD detection demonstrate the effectiveness of DCM. In selective classification, DCM consistently outperforms 6 representative approaches in conditions of distribution shift by 2.3% across 4 distribution-shift datasets. DCM also outperforms an ensemble of 5 models on 3 out of 4 datasets in AUROC, despite the 5× difference in computational cost. In the OOD detection setting, among other methods, we provide a comparison with Outlier Exposure (Hendrycks et al., 2018), allowing us to test our choice of uncertainty dataset. DCM consistently outperforms Outlier Exposure on a benchmark of 8 ID-OOD distribution pairs, reducing FPR (at TPR 95%) by 6.3% and 58.1% on CIFAR-10 and CIFAR-100, respectively. DCM also shows strong performance in challenging near-OOD detection settings, achieving 1.89% and 2.94% higher AUROC compared to the state-of-the-art.

## 2 Problem Setup

We consider two problem settings that test a model's ability to determine if its prediction is trustworthy: selective classification and out-of-distribution (OOD) detection. In both settings, a model may encounter *known* or *unknown* examples at test time. *Known* examples are inputs that are well-represented in the training distribution; *unknown* examples are not.

We denote the input and label spaces as $\mathcal{X}$ and $\mathcal{Y}$, respectively, and assume that the training dataset $D_{\mathrm{tr}}$ contains known examples. In *selective classification*, all inputs have a ground-truth label within $\mathcal{Y}$, but the model may make errors due to overfitting or insufficient coverage in the training dataset $D_{\mathrm{tr}}$. In this setting, known examples are inputs that are correctly classified by the model and unknown examples are inputs which are misclassified. In *out-of-distribution detection*, the model may encounter inputs at test time that belong to a new class not in its training label space $\mathcal{Y}$. In this setting, known examples are those from the training input space $\mathcal{X}$, and unknown examples are inputs from novel classes. We first describe these problem settings in Section 2.1 and Section 2.2. In Section 3, we present two instantiations of DCM for selective classification and OOD detection.

### 2.1 Selective Classification

Selective classification aims to produce a model that can abstain from making predictions on unknown examples at test time. Such "rejected" inputs are typically ones that the model is most uncertain about. A model is first trained on $D_{\mathrm{tr}}$ and then tested on examples that have associated ground-truth labels in the training label space $\mathcal{Y}$. Thus, while a perfect model trained on $D_{\mathrm{tr}}$ should correctly classify all test inputs, models often make errors on new examples due to over-fitting $D_{\mathrm{tr}}$. As in prior work, we assume access to a labeled validation dataset $D_{\mathrm{val}}$ sampled from $\mathcal{P}_{\mathrm{ID}}$, which the model can use to calibrate its predictive confidence. This validation set can be easily constructed by randomly partitioning a training dataset into training and validation splits.

Models are evaluated on their ability to (1) accurately classify the inputs they do make a prediction on (i.e., accuracy), while (2) rejecting as few inputs as possible (i.e., coverage). The confidence threshold is chosen to achieve a desired accuracy or coverage based on the specific application's risk tolerance, which can be done using a held-out validation set. We evaluate these capabilities through metrics such as ECE, AUC, Acc@Cov, Cov@Acc. Section 6 describes these metrics and the datasets we use.

### 2.2 Out-of-Distribution Detection

Out-of-distribution detection aims to distinguish between known and unknown examples at test time. We denote the ID distribution with $\mathcal{P}_{\mathrm{ID}}$, and the OOD distribution with $\mathcal{P}_{\mathrm{OOD}}$. The test dataset is a mixture of known and unknown examples, sampled from a mixture of the ID and OOD distributions $\alpha_{\mathrm{test}}\mathcal{P}_{\mathrm{ID}} + (1 - \alpha_{\mathrm{test}})\mathcal{P}_{\mathrm{OOD}}$, where the mixture coefficient $\alpha_{\mathrm{test}}$ is not known in advance. Given a test input $x$, the model produces both a label prediction and a measure of confidence that should be higher for inputs

that are known, or ID, than inputs that are unknown, or OOD. Due to differences in the data distributions $\mathcal{P}_{\text{ID}}$ and $\mathcal{P}_{\text{OOD}}$, a model trained solely to minimize loss on $D_{\text{tr}}$ may be overconfident on unknown inputs. To address this challenge, we use an additional unlabeled dataset $D_{\text{u}}$ which includes both known and unknown examples. $D_{\text{u}}$ is sampled from a mixture of $\mathcal{P}_{\text{ID}}$ and $\mathcal{P}_{\text{OOD}}$, where the mixture ratio $\alpha_{\text{u}}$ is unknown to the model and can differ from $\alpha_{\text{test}}$. The unlabeled dataset partially informs the model about the directions of variation it may face at test time.

Models are evaluated on (1) their accuracy in classifying known examples, and (2) their capability to detect unknown examples. These are measured by metrics such as FPR@TPR, AUROC, AUPR, and ID Accuracy. Section 6 describes these metrics and the ID-OOD dataset pairs we use.

## 3    Data-Driven Confidence Minimization

We aim to produce a model that achieves high accuracy on the training data $D_{\text{tr}}$, while having a predictive confidence that reflects the extent to which its prediction can be trusted. The crux of DCM is to introduce a regularizer that minimizes confidence on an auxiliary dataset that is disjoint from the training dataset. We refer to this auxiliary dataset as the *uncertainty dataset*, since it is intended to at least partly consist of examples that the model should have low confidence on.

In DCM, we first pre-train a model $f : \mathcal{X} \to \mathbb{P}(\mathcal{Y})$ on the labeled training set using the standard cross-entropy loss, as in prior works (Hendrycks et al., 2018; Liu et al., 2020):

$$\mathcal{L}_{\text{xent}}(f, D_{\text{tr}}) = \mathop{\mathbb{E}}_{(x,y) \sim D_{\text{tr}}} \left[ -\log f(y; x) \right]. \tag{1}$$

A model trained solely with this loss can suffer from overconfidence on unknown examples. We therefore continue to fine-tune this model on known examples, but simultaneously regularize to minimize the model's predictive confidence on an uncertainty dataset, which includes unknown examples. Specifically, we minimize cross-entropy loss on a fine-tuning dataset of known examples that includes the original training data ($D_{\text{tr}} \subseteq D_{\text{ft}}$). Our additional regularizer minimizes confidence on the uncertainty dataset $D_{\text{unc}}$:

$$\mathcal{L}_{\text{conf}}(f, D_{\text{unc}}) = \mathop{\mathbb{E}}_{x' \sim D_{\text{unc}}} \left[ -\log f(y_u; x') \right]. \tag{2}$$

Here, $y_u$ is a uniform target that assigns equal probability to all labels. The confidence loss $\mathcal{L}_{\text{conf}}$ is equivalent to the KL divergence between the predicted probabilities and the uniform distribution $U$. Our final objective is a weighted sum of the fine-tuning and confidence losses:

$$\mathcal{L}_{\text{xent}}(f, D_{\text{ft}}) + \lambda \mathcal{L}_{\text{conf}}(f, D_{\text{unc}}), \tag{3}$$

where $\lambda$ is a hyperparameter that controls the relative weight of the confidence loss term. We find that $\lambda = 0.5$ works well in practice and use this value in all experiments unless otherwise specified. Further details, such as fine-tuning duration and the number of samples in $D_{\text{ft}}$ and $D_{\text{unc}}$, are described in Appendix C. In our experiments, we find that using an uncertainty set that is around 10% of the training set size is sufficient.

The two instantiations of DCM for OOD detection and selective classification differ only in their construction of $D_{\text{ft}}$ and $D_{\text{unc}}$, as we will describe in Section 3.2 and Section 3.1.

### 3.1    DCM for Selective Classification

We aim to produce a model that achieves high accuracy while having low confidence on inputs that it is likely to misclassify. We expect the incorrect predictions of a model $f$ pretrained on $D_{\text{tr}}$ to reflect its failure modes. Recall from Section 2.1 that we assume a held-out validation set $D_{\text{val}} \sim \mathcal{P}_{\text{ID}}$. To better calibrate its predictive confidence, we compare our pre-trained model's predictions for inputs in $D_{\text{val}}$ to their ground-truth labels, and obtain the set of correct and misclassified validation examples $D_{\text{val}}^{\circ}, D_{\text{val}}^{\times}$. In this setting, the unknown examples are the misclassified examples $D_{\text{val}}^{\times}$, since they show where the model's learned decision boundary is incorrect.

---

**Algorithm 1** DCM for Selective Classification

> **Input:** Training data $D_{\text{tr}}$, Validation data $D_{\text{val}}$, Hyperparameter $\lambda$
>
> Initialize weights $\theta \leftarrow \theta_0$
> **while** Not converged **do**
>    Sample mini-batch $B_{\text{tr}} \sim D_{\text{tr}}$
>    Update $\theta$ using $\nabla_\theta \mathcal{L}_{\text{xent}}(B_{\text{tr}}, f)$
> **end while**
> Get correct set $D_{\text{val}}^\circ \leftarrow \{(x,y) \in D_{\text{val}} \mid f_\theta(x) = y\}$
> Get error set $D_{\text{val}}^\times \leftarrow \{(x,y) \in D_{\text{val}} \mid f_\theta(x) \neq y\}$
> **while** Not converged **do**
>    Sample mini-batches $B_{\text{tr}} \sim D_{\text{tr}} \cup D_{\text{val}}^\circ, B_{\text{val}}^\times \sim D_{\text{val}}^\times$
>    Update $\theta$ using $\nabla_\theta \mathcal{L}_{\text{xent}}(B_{\text{tr}}, f) + \lambda \mathcal{L}_{\text{conf}}(B_{\text{val}}^\times, f)$
> **end while**

**Algorithm 2** DCM for OOD Detection

> **Input:** Training data $D_{\text{tr}}$, Unlabeled data $D_{\text{u}}$, Hyperparameter $\lambda$
>
> Initialize weights $\theta \leftarrow \theta_0$
> **while** Not converged **do**
>    Sample mini-batch $B_{\text{tr}} \sim D_{\text{tr}}$
>    Update $\theta$ using $\nabla_\theta \mathcal{L}_{\text{xent}}(f, B_{\text{tr}})$
> **end while**
> **while** Not converged **do**
>    Sample mini-batches $B_{\text{tr}} \sim D_{\text{tr}}, B_{\text{u}} \sim D_{\text{u}}$
>    Update $\theta$ using $\nabla_\theta \mathcal{L}_{\text{xent}}(f, B_{\text{tr}}) + \lambda \mathcal{L}_{\text{conf}}(f, B_{\text{u}})$
> **end while**

---

We set the fine-tuning dataset to be the union of the training dataset and the correctly-classified validation examples ($D_{\text{ft}} = D_{\text{tr}} \cup D_{\text{val}}^\circ$), and use the misclassified validation examples as the uncertainty dataset ($D_{\text{unc}} = D_{\text{val}}^\times$). By only minimizing confidence on the misclassified examples, we expect the model to have lower confidence on all examples similar to inputs which initially produced errors. We outline our approach in Algorithm 1.

### 3.2 DCM for Out-of-Distribution Detection

We aim to produce a model that has low confidence on unknown inputs from the OOD distribution $\mathcal{P}_{\text{OOD}}$, while achieving high accuracy on known inputs from the ID distribution $\mathcal{P}_{\text{ID}}$. Recall from Section 2.2 that our problem setting assumes access to an unlabeled dataset $D_{\text{u}}$, which includes both ID and OOD inputs: we use this unlabeled set as the uncertainty dataset for reducing confidence ($D_{\text{unc}} = D_{\text{u}}$). Intuitively, minimizing confidence on $D_{\text{u}}$ discourages the model from making overly confident predictions on the support of the uncertainty dataset.

We minimize confidence on all inputs in $D_{\text{u}}$ because it is not known a priori which inputs are ID versus OOD, or in our terminology, known versus unknown. While we do not necessarily want to minimize confidence on known inputs, confidence minimization is expected to have different effects on known and unknown inputs because of its interaction with the original cross-entropy loss. On known inputs, the effect of confidence minimization is "cancelled out" by the cross-entropy loss, which maximizes the log likelihood of the true label, thus increasing the predictive confidence for that input. However, on unknown inputs, the loss is solely confidence minimization, which forces the model to have low confidence on such inputs. This allows DCM to differentiate between the ID and OOD data distributions based on predictive confidence. We will formalize this intuition in Section 4. In summary, in OOD detection, the fine-tuning dataset is the training dataset ($D_{\text{ft}} = D_{\text{tr}}$), and the uncertainty dataset is the unlabeled dataset ($D_{\text{unc}} = D_{\text{u}}$). We outline our approach in Algorithm 2.

## 4 Analysis

We now theoretically analyze the effect of the DCM objective on known and unknown inputs. We first show that for all test examples, the prediction confidence of DCM is a lower bound on the true confidence (Proposition 4.1). Using this property, we then demonstrate that DCM can provably detect unknown examples similar to those in the uncertainty set with an appropriate threshold on predicted confidence (Proposition 4.2). Detailed statements and proofs can be found in Appendix A.

We denote the true label distribution of input $x$ as $p_D(x)$; this distribution need not be a point mass on a single label. We further denote the maximum softmax probability of any distribution $p$ as $\text{MSP}(p) \triangleq \max_i p_i$, and denote by $f_\lambda(x)$ the predictive distribution of the model that minimizes the expectation of our objective (3) with respect to the data distribution $y \sim p_D(x)$ for input $x$. Intuitively, the confidence minimization term in

our objective function (3) forces the model to output low-confidence predictions on all datapoints, resulting in a more conservative model compared to one without this term. We formalize this intuition in the following proposition which relates the maximum softmax probabilities of $f_\lambda$ and $p_D$.

**Proposition 4.1** (Lower bound on true confidence). *For any input $x$ in $D_u$ or $D_{tr}$, the optimal predictive distribution $f_\lambda$ satisfies $MSP(f_\lambda) \leq MSP(p_D)$, with equality if and only if $\lambda = 0$.*

We note that this proposition only assumes a sufficiently expressive model class, which large neural networks often are.

Now we restrict ourselves to using an unlabeled mixture of known and unknown examples, $D_{\mathrm{u}}$, as the uncertainty set. Beyond serving as a lower bound on the true confidence, the optimum distribution $p_\lambda$ shows how the model, after being fine-tuned to minimize confidence on the unlabeled dataset $D_{\mathrm{u}}$, behaves differently for known and unknown data despite $D_{\mathrm{u}}$ containing both. We denote the subset of known examples in $D_{\mathrm{u}}$ as $D_k^{test}$, the unknown subset as $D_{unk}^{test}$, and the $\delta$-neighborhoods of these two sets as $D_k^\delta, D_{unk}^\delta$; we give a precise definition in Appendix A. For ID inputs, the optimal predictive distribution $p_\lambda$ is determined by the weighted sum of the cross-entropy loss and the confidence loss, resulting in a mixture between the true label distribution $p$ and the uniform distribution $\mathcal{U}$, with mixture weight $\lambda$. On the other hand, for unknown inputs, the confidence loss is the only loss term, thus the optimal predictive distribution $p_\lambda$ is the uniform distribution $\mathcal{U}$. This distinct behavior allows for the detection of unknown inputs by thresholding the confidence of the model's predictions, as formalized in the following proposition.

**Proposition 4.2** (Low loss implies separation). *Assume $D_k^\delta$ and $D_{unk}^\delta$ are disjoint, and that each input $x$ has only one ground-truth label, i.e., no label noise. Denote the lowest achievable loss for the objective 3 with $\lambda > 0$ as $\mathcal{L}_0$. Under a mild smoothness assumption on the learned function $f_\theta$, there exists $\epsilon, \delta > 0$ such that $\mathcal{L}(\theta) - \mathcal{L}_0 < \epsilon$ implies the following relationship between the max probabilities:*

$$\inf_{x \in D_k^\delta} MSP(f_\theta(x)) > \sup_{x \in D_{unk}^\delta} MSP(f_\theta(x)). \tag{4}$$

The detailed smoothness assumption, along with all proofs, can be found in Appendix A. This proposition implies that by minimizing the DCM objective (3), we can provably separate out known and unknown data with an appropriate threshold on the maximum softmax probability. We note that DCM optimizes a lower bound on confidence, rather than trying to be perfectly calibrated: this easier requirement is arguably better suited for problem settings in which the model abstains from making predictions such as OOD detection and selective classification. When fine-tuning the last layers of a pre-trained network, Proposition 4.2 directly applies to feature-space distances, which are known to reflect semantic relations Upchurch et al. (2017); Wang et al. (2019).

**Practical implications.** Our theory suggests the following guidelines. First, the uncertainty dataset should include some unknown examples. Second, if this is true, it is unnecessary to filter out known examples for OOD detection. Under these conditions, DCM provably detects unknown examples.

## 5 Related Work

**Selective classification.** Prior works have studied selective classification for many model classes including SVM, boosting, and nearest neighbors (Hellman, 1970; Fumera & Roli, 2002; Cortes et al., 2016). Because deep neural networks generalize well but are often overconfident (Guo et al., 2017; Nixon et al., 2019), mitigating such overconfidence using selective classification while preserving its generalization properties is an important capability (Geifman & El-Yaniv, 2017; Corbière et al., 2019; Feng et al., 2019; Kamath et al., 2020; Fisch et al., 2022). Existing methods for learning conservative neural networks rely on additional assumptions such as pseudo-labeling (Chan et al., 2020), multiple distinct validation sets (Gangrade et al., 2021), or adversarial OOD examples (Setlur et al., 2022). While minimizing the confidence of a set that includes OOD inputs has been shown to result in a more conservative model in the offline reinforcement learning setting (Kumar et al., 2020), this approach has not been validated in a supervised learning setting. DCM only requires a small validation set, and our experiments in Section 6 show that it performs competitively with

state-of-the-art methods for selective classification, especially in the more challenging setting of distribution shift.

**Out-of-distribution detection.** Many existing methods for OOD detection use a criterion based on the activations or predictions of a model trained on ID data (Bendale & Boult, 2016; Hendrycks & Gimpel, 2016; Liang et al., 2017b; Lee et al., 2018; Wei et al., 2022; Sun et al., 2022). However, the performance of these methods are often inconsistent across different ID-OOD dataset pairs, suggesting that the OOD detection problem is ill-defined (Tajwar et al., 2021). Indeed, a separate line of work incorporates auxiliary data into the OOD detection setting; this dataset may consist of natural (Hendrycks et al., 2018; Liu et al., 2020; Mohseni et al., 2020; Ţifrea et al., 2020; Chen et al., 2021; Katz-Samuels et al., 2022; Narasimhan et al., 2023) or synthetic (Lee et al., 2017; Du et al., 2022b) data. Similar to our method, Hendrycks et al. (2018) minimize confidence on an auxiliary dataset, but do so on a single auxiliary dataset of known outliers, regardless of the ID and OOD distributions, that is over $10,000$ times the size of those used by DCM. Our method leverages an uncertainty dataset which contains a mix of ID and OOD data from the test distribution, as in Ţifrea et al. (2020). However, their method requires an ensemble of models to measure disagreement, while DCM uses a single model. We additionally present theoretical results showing the benefit of minimizing confidence on an uncertainty dataset that includes inputs from the OOD distribution. Our experiments confirm our theory, showing that this transductive setting results in substantial performance gains, even when the unlabeled set is a mixture of ID and OOD data. Our data assumptions are also shared by WOODS (Katz-Samuels et al., 2022), which leverages an auxiliary dataset containing ID and OOD examples. However, WOODS solves a constrained optimization problem to maximize OOD detection rate while keeping ID classification and OOD error rate of ID examples low, which requires several additional hyperparameters compared to DCM, which uses confidence minimization.

**Data augmentation.** Data augmentation is crucial for enhancing the robustness of models, particularly in uncertainty quantification and OOD detection. By generating diverse examples, augmentation exposes the model to a wider range of variations, improving generalization and OOD detection. Hafner et al. (2020) showed that synthetic augmentation significantly boosts OOD performance. Similarly, Hendrycks et al. (2018) utilized various data augmentation techniques to enhance model robustness against OOD inputs. Integrating such strategies with our framework could enhance its effectiveness by providing a richer set of uncertainty data, improving the model's ability to manage uncertainty.

Recent works highlight the need for an integrated approach to handle both ID misclassifications and OOD samples effectively. Jaeger et al. (2022) show that existing methods often fail to address all relevant error types simultaneously. In this vein, Xia & Bouganis (2022) propose a method to distinguish between correctly and incorrectly classified ID samples using softmax-based confidence scores, while also detecting OOD samples. Future work could extend DCM to this problem setting.

# 6 Experiments

We evaluate the effectiveness of DCM for selective classification and OOD detection using several image classification datasets. Our goal is to empirically answer the following questions: (1) How does the data-driven confidence minimization loss affect the predictive confidence of the final model, and what role does the distribution of the uncertainty data play? (2) Does confidence minimization on the uncertainty dataset result in better calibration? (3) How does DCM compare to state-of-the-art methods for OOD detection and selective classification?

**Metrics.** Recall that the selective classification problem involves a binary classification task to predict whether the model will misclassify a given example, in addition to the main classification task. Similarly, the OOD detection problem involves two classification tasks: (1) a binary classification task to predict whether each example is ID or OOD, and (2) the main classification task of predicting labels of images. To ensure a comprehensive evaluation, we consider multiple metrics, each measuring the two key aspects of performance. We group the metrics below by their relevance to the selective classification (SC) or OOD detection (D) setting. These metrics are defined in detail in Appendix B:

1. ECE (SC): expected difference between confidence and accuracy, i.e., $\mathbb{E}[|p(\hat{y} = y \mid \hat{p} = p) - p|]$.

| Setting | Method | CIFAR-10 | | | Waterbirds | | | Camelyon17 | | | FMoW | | |
|---|---|---|---|---|---|---|---|---|---|---|---|---|---|
| | | Acc (↑) | Acc@90 (↑) | AUC (↑) | Acc (↑) | Acc@90 (↑) | AUC (↑) | Acc (↑) | Acc@90 (↑) | AUC (↑) | Acc (↑) | Acc@90 (↑) | AUC (↑) |
| ID | Ensemble (×5) | **96.1 (0.1)\*** | **98.9 (0.1)\*** | **99.5 (0.1)\*** | **97.0 (0.0)\*** | **98.9 (0.0)\*** | **98.7 (0.0)\*** | 94.8 (6.4)* | 96.8 (5.9)* | 99.1 (2.7)* | **62.5 (0.1)\*** | **68.4 (0.1)\*** | **85.5 (0.0)\*** |
| | MSP | 95.2 (0.1) | 98.4 (0.1) | 99.3 (0.1) | 96.8 (0.0) | 99.1 (0.0) | **98.7 (0.0)** | 81.5 (7.8) | 92.0 (5.9) | 96.9 (2.2) | 58.4 (1.5) | 62.6 (0.1) | 81.3 (0.4) |
| | MaxLogit | 95.1 (0.1) | 97.9 (0.1) | 98.9 (0.1) | 96.8 (0.0) | 97.2 (0.0) | 98.6 (0.0) | 81.5 (7.8) | 92.2 (5.8) | 97.0 (2.2) | 58.4 (0.1) | 62.7 (0.2) | 80.1 (0.2) |
| | Binary Classifier | 95.2 (0.1) | 98.4 (0.1) | 99.3 (0.1) | 96.0 (0.0) | 99.1 (0.0) | **98.7 (0.0)** | 89.4 (6.5) | 92.3 (5.9) | 97.0 (4.5) | 58.4 (0.2) | 64.3 (0.1) | 82.3 (0.3) |
| | Fine-Tuning | **96.2 (0.1)** | **99.1 (0.2)** | **99.6 (0.0)** | 96.9 (0.0) | **99.4 (0.0)** | **98.7 (0.0)** | **98.3 (0.2)** | **99.7 (0.0)** | **99.8 (0.0)** | 59.3 (2.7) | 64.0 (1.2) | 82.8 (0.9) |
| | Deep Gamblers | 94.5 (0.0) | 97.4 (0.1) | 99.0 (0.0) | **97.0 (0.0)** | 98.8 (0.1) | 98.5 (0.0) | 97.5 (0.4) | **99.6 (0.1)** | **99.8 (0.0)** | 58.5 (0.4) | 62.4 (0.9) | 75.8 (0.2) |
| | Self-Adaptive Training | 94.7 (0.0) | 97.6 (0.1) | 99.2 (0.0) | 96.8 (0.0) | 99.1 (0.1) | 98.6 (0.0) | **97.7 (0.0)** | **99.7 (0.0)** | **99.8 (0.0)** | 58.3 (0.5) | 63.0 (0.5) | 81.1 (0.3) |
| | DCM (ours) | 94.7 (0.2) | 98.0 (0.2) | 99.2 (0.0) | 96.8 (0.0) | 99.2 (0.0) | **98.7 (0.0)** | 80.6 (1.0) | 98.6 (0.2) | 99.5 (0.1) | 59.3 (1.2) | 64.2 (1.2) | 82.9 (1.1) |
| ID + OOD | Ensemble (×5) | **76.4 (0.1)\*** | **81.2 (0.1)\*** | **93.4 (0.1)\*** | – | – | – | 75.6 (4.6)* | 78.1 (4.8)* | 85.8 (3.7)* | **56.5 (0.0)\*** | **61.2 (0.0)\*** | **81.7 (0.0)\*** |
| | MSP | 75.8 (0.1) | 80.3 (0.1) | 92.6 (0.1) | – | – | – | 66.2 (5.1) | 74.1 (5.1) | 72.2 (4.8) | 51.5 (0.1) | 57.9 (0.1) | 77.1 (0.5) |
| | MaxLogit | 75.7 (0.1) | 80.4 (0.0) | 91.7 (0.0) | – | – | – | 66.2 (5.1) | 74.2 (5.1) | 85.8 (3.7) | 51.5 (0.1) | 57.8 (0.1) | 75.8 (0.1) |
| | Binary Classifier | 75.4 (0.1) | 80.3 (0.1) | 92.5 (0.1) | – | – | – | 86.2 (3.3) | 74.4 (5.0) | 72.0 (4.7) | 53.8 (0.1) | 59.3 (0.0) | 78.0 (0.4) |
| | Fine-Tuning | 75.2 (0.1) | 81.3 (0.1) | 93.4 (0.1) | – | – | – | 76.7 (3.4) | 79.8 (3.5) | 77.6 (3.3) | 54.2 (2.3) | 58.6 (1.2) | 78.6 (0.8) |
| | Deep Gamblers | 76.0 (0.1) | 81.0 (0.0) | 93.0 (0.1) | – | – | – | 74.0 (5.8) | 77.2 (6.5) | 88.1 (4.1) | 54.0 (0.3) | 57.5 (0.3) | 71.6 (0.2) |
| | Self-Adaptive Training | 76.2 (0.1) | 81.1 (0.0) | 93.3 (0.0) | – | – | – | 72.1 (1.1) | 74.8 (1.1) | 86.3 (0.4) | 53.7 (0.4) | 57.8 (0.4) | 76.7 (0.2) |
| | DCM (ours) | **77.0 (0.1)** | **82.0 (0.1)** | **93.6 (0.1)** | – | – | – | **80.6 (1.0)** | **85.5 (1.0)** | **93.5 (0.6)** | 54.6 (1.7) | 58.8 (1.3) | 78.9 (1.1) |
| OOD | Ensemble (×5) | 57.2 (0.1)* | 61.8 (0.1)* | 75.3 (0.1)* | 85.0 (0.0)* | 88.4 (0.0)* | 94.4 (0.0)* | 71.8 (4.8)* | 74.0 (5.2)* | 81.4 (4.4)* | **55.0 (0.1)\*** | **58.6 (0.1)\*** | **79.5 (0.0)\*** |
| | MSP | 56.4 (0.1) | 59.6 (0.2) | 70.1 (0.1) | 84.3 (0.0) | 88.2 (0.0) | 94.4 (0.0) | 63.1 (4.8) | 70.4 (4.8) | 82.2 (3.9) | 50.9 (2.7) | 55.2 (0.2) | 74.5 (0.6) |
| | MaxLogit | 56.4 (0.1) | 59.4 (0.1) | 71.7 (0.1) | 84.3 (0.0) | 87.9 (0.0) | 94.2 (0.0) | 63.1 (4.8) | 70.4 (4.8) | 82.1 (3.9) | 50.7 (0.1) | 55.2 (0.0) | 73.3 (0.2) |
| | Binary Classifier | 56.2 (0.2) | 59.5 (0.2) | 72.8 (0.2) | 84.9 (0.2) | 87.5 (0.3) | 94.0 (0.2) | 69.0 (5.2) | 70.5 (4.4) | 82.4 (3.9) | 51.7 (0.0) | 56.8 (0.1) | 75.6 (0.5) |
| | Fine-Tuning | 57.6 (0.4) | 61.9 (0.2) | 75.4 (0.1) | 85.9 (0.5) | 89.0 (0.5) | 94.7 (0.2) | 72.8 (4.2) | 75.4 (4.2) | 84.2 (3.8) | 51.8 (1.1) | 56.0 (0.9) | 76.2 (0.8) |
| | Deep Gamblers | 56.8 (0.1) | 61.4 (0.1) | 74.3 (0.2) | 85.1 (0.1) | 88.6 (0.2) | 94.8 (0.1) | 69.4 (7.5) | 72.1 (7.9) | 84.8 (5.2) | 51.9 (0.1) | 54.9 (0.2) | 69.2 (0.3) |
| | Self-Adaptive Training | 57.4 (0.1) | 61.4 (0.1) | 75.3 (0.1) | 86.0 (0.0) | 88.9 (0.1) | 95.1 (0.0) | 70.2 (0.7) | 71.9 (0.8) | 80.3 (0.6) | 51.0 (0.4) | 55.1 (0.4) | 74.1 (0.2) |
| | DCM (ours) | **59.4 (0.1)** | **64.1 (0.2)** | **77.5 (0.2)** | **86.5 (0.2)** | **89.5 (0.3)** | 95.0 (0.1) | **78.7 (1.2)** | **82.5 (1.2)** | **91.6 (1.1)** | 51.9 (1.7) | 56.2 (1.4) | 76.4 (1.1) |

\* Ensemble requires 5× the compute compared to other methods.

Table 1: Selective classification performance on four datasets. Numbers in parentheses represent the standard error over 3 seeds, and we bold all methods that have overlapping error with the best-performing method. DCM consistently achieves the best performance in settings with distribution shift (ID+OOD, OOD).

2. AUC (SC): area under the curve of selective classification accuracy vs coverage.

3. Acc@Cov (SC): average accuracy on the Cov% datapoints with highest confidence.

4. Cov@Acc (SC): largest fraction of data for which selective accuracy is above Acc.

5. FPR@TPR (D): probability that an ID input is misclassified as OOD, given true positive rate TPR.

6. AUROC (D): area under the receiver operator curve of the binary ID/OOD classification task.

7. AUPR (D): area under the precision-recall curve of the binary ID/OOD classification task.

## 6.1 Selective Classification

We assess the capability of models fine-tuned with DCM to abstain from making incorrect predictions. We evaluate on several toy and real-world image classification datasets that exhibit distribution shift.

**Datasets.** We evaluate selective classification performance on CIFAR-10 (Krizhevsky et al., a) and CIFAR-10-C (Hendrycks & Dietterich, 2019), Waterbirds (Sagawa et al., 2019; Wah et al., 2011), Camelyon17 (Koh et al., 2021), and FMoW (Koh et al., 2021). These datasets were chosen to evaluate selective classification performance in the presence of diverse distribution shifts: corrupted inputs in CIFAR-10-C, spurious correlations in Waterbirds, and new domains in Camelyon17 and FMoW.

The ID/OOD/ID+OOD settings for each dataset are constructed as follows. For CIFAR-10, the ID dataset is CIFAR-10, the OOD dataset is CIFAR-10-C, and the ID+OOD dataset is a 50-50 mix of the two datasets. For Waterbirds, the ID dataset is the training split and the OOD dataset is a group-balanced validation set; we do not consider an ID+OOD dataset here. For Camelyon17, the ID dataset consists of images from the first 3 hospitals, which are represented in the training data. The OOD dataset consists of images from the last 2 hospitals, which do not appear in the training set. The ID+OOD dataset is a mix of all five hospitals. For FMoW, the ID dataset consists of images collected from the years $2002 - 2013$, which are represented in the training data. The OOD setting tests on images collected between $2016 - 2018$, and the ID + OOD setting tests on images from $2002 - 2013$ and $2016 - 2018$.

**Comparisons.** We consider 7 representative prior methods as baselines: MSP (Hendrycks & Gimpel, 2016), MaxLogit (Hendrycks et al., 2022), Binary Classifier (Kamath et al., 2020), Fine-Tuning on the labeled ID validation set, Deep Gamblers (Liu et al., 2019), and Self-Adaptive Training (Huang et al., 2020), and an ensemble of 5 MSP models as a rough upper bound on performance given more compute. All methods use the same ID validation set for hyperparameter tuning and/or calibration.

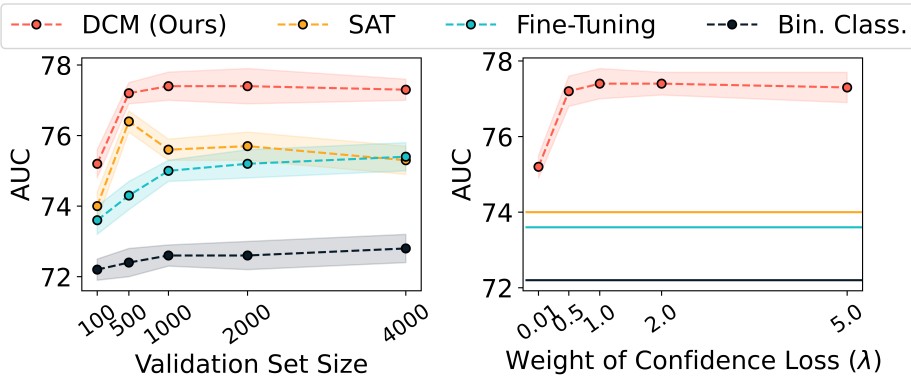

Figure 2: Selective classification performance of DCM on the CIFAR-10 → CIFAR-10-C task with validation set sizes (left) and various confidence loss weights $\lambda$ (right).

**DCM outperforms prior methods when testing on unseen distributions.** We present representative metrics in Table 1 and full metrics in Tables 15 to 17. DCM consistently outperforms all baselines in settings of distribution shift (OOD and ID+OOD). These performance gains are attributed to DCM's confidence minimization on uncertain examples, which better prepares the model for unfamiliar inputs. DCM even outperforms Ensemble on three of the four datasets, despite requiring 1/5 of the compute. Fine-Tuning outperforms DCM when the training and validation datasets are from the same distribution (ID). In settings where the test distribution differs from the training and validation distributions, DCM outperforms Fine-Tuning on most metrics. These experiments indicate that DCM learns a more conservative model in conditions of distribution shift, compared to state-of-the-art methods for selective classification.

**DCM is robust to a range of confidence loss weights, $\lambda$, and validation set sizes.** We investigate the sensitivity of DCM to the size of the validation set in Figure 2 (left). We find that DCM for selective classification is robust to a range of validation set sizes. This robustness suggests that DCM effectively leverages the available validation data to calibrate its confidence, maintaining performance even with varying amounts of data. In Figure 2 (right), we plot the performance of DCM with various values of $\lambda$ on tasks constructed from the CIFAR-10 and CIFAR-10-C datasets. We find that DCM performs best with $\lambda = 0.5$, indicating that this balance effectively regularizes the model without over-penalizing confident predictions. This balance allows DCM to maintain high accuracy while ensuring conservative prediction on uncertain inputs.

## 6.2 OOD Detection

We evaluate DCM on the standard OOD detection setting and the more challenging near-OOD detection setting. We evaluate three variants of DCM, each using the training objective described in Section 3, but with three different measures of confidence: MSP (Hendrycks & Gimpel, 2016), MaxLogit (Hendrycks et al., 2022), and Energy (Liu et al., 2020). We denote these three variants as DCM-Softmax, DCM-MaxLogit, DCM-Energy, and describe these variants in detail in Appendix C. All experiments in this section use $\lambda = 0.5$ and the default hyperparameters from Hendrycks et al. (2018). Further experimental details are in Appendix D.

**Datasets.** We use CIFAR-10 and CIFAR-100 as our ID datasets and TinyImageNet, LSUN, iSUN and SVHN as our OOD datasets, resulting in a total of 8 ID-OOD pairs. We split the ID data into 40,000 examples for training and 10,000 examples for validation. Our uncertainty and test sets are disjoint datasets with 5,000 and 1,000 examples, respectively. On the near-OOD detection tasks, the ID and OOD datasets consist of disjoint classes in the same dataset. The number of examples per class is the same as in the standard OOD detection setting. For comparison on large-scale image datasets, we use ImageNet-1K as ID and iNaturalist, SUN, Textures and Places as OOD datasets. Please refer to Appendix I for further experimental details and the full comparison table.

| | | ID Dataset | | | | | |
| | | CIFAR-10 | | | CIFAR-100 | | |
| Method | Architecture | ID Acc (↑) | AUROC (↑) | FPR@95 (↓) | ID Acc (↑) | AUROC (↑) | FPR@95 (↓) |
|---|---|---|---|---|---|---|---|
| MSP | | 94.7 | 90.7 | 30.9 | 73.9 | 70.3 | 72.1 |
| Energy Score | | 94.7 | 91.5 | 33.2 | 73.9 | 76.0 | 66.7 |
| MaxLogit | | 94.7 | 93.5 | 25.5 | 73.9 | 77.7 | 63.5 |
| ODIN | | 94.7 | 94.6 | 24.1 | 73.9 | 84.5 | 53.0 |
| Mahalanobis | | 94.7 | 93.6 | 27.3 | 73.9 | 91.6 | 37.3 |
| Outlier Exposure | WRN-40-2 | 94.7 | 98.5 | 6.6 | **75.7** | 81.1 | 59.4 |
| Energy Fine-Tuning | | **95.1** | 99.1 | 3.4 | 75.2 | 81.5 | 59.6 |
| WOODS | | 94.7 | 94.8 | 4.1 | 72.7 | 98.0 | 12.9 |
| DCM-Softmax (ours) | | 93.6 | 99.6 | 1.0 | 71.2 | 99.2 | 2.6 |
| DCM-MaxLogit (ours) | | 93.6 | **99.8** | 0.7 | 71.2 | 99.4 | 1.7 |
| DCM-Energy (ours) | | 93.6 | 99.7 | **0.3** | 71.2 | **99.5** | **1.3** |
| Binary Classifier | | - | 98.9 | 1.3 | - | 97.9 | 7.6 |
| ERD | | - | 99.5* | 1.0* | - | 99.1* | 2.6* |
| DCM-Softmax (ours) | ResNet-18 | **93.4** | 99.5 | 1.9 | 70.9 | 99.1 | 4.6 |
| DCM-MaxLogit (ours) | | **93.4** | 99.5 | 1.5 | 70.9 | 99.2 | 3.5 |
| DCM-Energy (ours) | | **93.4** | 99.5 | 1.4 | **71.0** | **99.3** | **2.3** |

\* ERD requires 3× the compute compared to other methods.

Table 2: OOD detection performance of models trained on CIFAR-10 or CIFAR-100 and evaluated on 4 OOD datasets. Metrics are averaged over OOD datasets; detailed dataset-specific results are in Table 7. The three variants of DCM exhibit competitive performance on all datasets.

| Methods | iNaturalist | | SUN | | Places | | Textures | | |
| | FPR@95 (↓) | AUROC (↑) | FPR@95 (↓) | AUROC (↑) | FPR@95 (↓) | AUROC (↑) | FPR@95 (↓) | AUROC (↑) | ID Acc (↑) |
|---|---|---|---|---|---|---|---|---|---|
| MCM (zero-shot) | 32.1 | 94.4 | 39.2 | 92.3 | 44.9 | 89.8 | 58.1 | 86.0 | 68.5 |
| MSP | 54.1 | 87.4 | 73.4 | 78.0 | 73.0 | 78.0 | 68.9 | 79.1 | **79.6** |
| ODIN | 30.2 | 94.7 | 54.0 | 87.2 | 55.1 | 85.5 | 51.7 | 87.9 | **79.6** |
| Energy | 29.8 | 94.7 | 53.2 | 87.3 | 56.4 | 85.6 | 51.4 | 88.0 | **79.6** |
| GradNorm | 81.5 | 72.6 | 82.0 | 72.9 | 80.4 | 73.7 | 79.4 | 70.3 | **79.6** |
| ViM | 32.2 | 93.2 | 54.0 | 87.2 | 60.7 | 83.8 | 53.9 | 87.2 | **79.6** |
| KNN | 29.2 | 94.5 | 35.6 | 92.7 | 39.6 | 91.0 | 64.4 | 85.7 | **79.6** |
| VOS | 31.7 | 94.5 | 43.0 | 91.9 | 41.6 | 90.2 | 56.7 | 86.7 | **79.6** |
| VOS+ | 29.0 | 94.6 | 36.9 | 92.6 | 38.4 | 91.2 | 61.0 | 86.3 | **79.6** |
| NPOS | 16.6 | 96.2 | 43.8 | 90.4 | 45.3 | 89.4 | 46.1 | 88.8 | 79.4 |
| DCM-Softmax | 2.6 | 99.2 | 32.9 | 94.2 | 35.9 | 93.8 | 11.2 | 97.9 | 78.9 |
| DCM-MaxLogit | 1.8 | 99.4 | 27.5 | 94.9 | 32.5 | 94.5 | 8.2 | 98.3 | 78.9 |
| DCM-Energy | **0.5** | **99.6** | **24.5** | **95.8** | **30.8** | **95.4** | **4.3** | **98.8** | 78.9 |

Table 3: OOD detection performance of ViT-B/16 with ImageNet-1K as the in-distribution training data.

**Comparisons.** In the standard OOD detection setting, we compare DCM with 8 representative OOD detection methods: MSP (Hendrycks & Gimpel, 2016), MaxLogit (Hendrycks et al., 2022), ODIN (Liang et al., 2017a), Mahalanobis (Lee et al., 2018), Energy Score (Liu et al., 2020), Outlier Exposure (Hendrycks et al., 2018), Energy Fine-Tuning (Liu et al., 2020), and WOODS (Katz-Samuels et al., 2022). In the more challenging near-OOD detection setting, standard methods such as Mahalanobis perform poorly (Ren et al., 2021), so we compare DCM with Binary Classifier and ERD (Tifrea et al., 2022), which attain state-of-the-art performance. Similar to DCM, these methods leverage an unlabeled dataset containing ID and OOD inputs. For experiments on ImageNet-1K, we compare to VOS (Du et al., 2022a) and NPOS (Tao et al., 2023), prior SOTA methods that use synthetic outliers.

**DCM outperforms prior methods.** As expected, DCM outperforms prior methods due to provable separation of ID and OOD inputs by predictive confidence. DCM outperforms all 8 prior methods across 8 ID-OOD dataset pairs, as shown in Table 2. For the standard OOD detection setting, we report full results in Appendix D. For near-OOD detection, Table 4 shows that DCM outperforms Binary Classifier, and performs similarly to ERD with only 1/3 of the compute. DCM scales well to larger tasks: Table 3 shows that DCM outperforms all prior approaches on ImageNet-1K as well.

| Setting | Method | FPR@95 (↓) | FPR@99 (↓) | AUROC (↑) | AUPR (↑) | ID Acc (↑) |
|---|---|---|---|---|---|---|
| ID = CIFAR-10 [0:5]
OOD = CIFAR-10 [5:10] | MSP | 78.6 (3.6) | 94.5 (2.8) | 75.7 (0.6) | 38.5 (1.7) | **94.9 (0.4)** |
| | MaxLogit | 79.2 (3.4) | 94.4 (2.0) | 75.5 (0.8) | 40.3 (1.5) | **94.9 (0.4)** |
| | Binary Classifier | 78.6 (1.5) | 94.0 (0.5) | 71.8 (1.1) | 79.5 (0.7) | - |
| | ERD | 72.5 (1.7) | 92.1 (0.8) | 79.3 (0.3) | **47.9 (1.6)** | - |
| | DCM-Softmax (ours) | **66.0 (2.6)** | **89.2 (1.0)** | 81.2 (0.3) | 45.7 (0.6) | 94.0 (0.5) |
| | DCM-MaxLogit (ours) | 67.6 (5.6) | 89.2 (2.2) | 81.3 (0.6) | 46.1 (1.4) | 94.0 (0.5) |
| | DCM-Energy (ours) | 67.3 (2.7) | 89.1 (0.9) | 81.4 (0.6) | 46.3 (0.7) | 94.0 (0.5) |
| ID = CIFAR-100 [0:50]
OOD = CIFAR-100 [50:100] | MSP | 68.8 (1.2) | 90.9 (1.1) | 75.4 (0.8) | **33.4 (1.4)** | **78.3 (0.5)** |
| | MaxLogit | 69.8 (2.0) | 91.5 (1.7) | **76.0 (0.5)** | **33.9 (1.4)** | **78.3 (0.5)** |
| | Binary Classifier | 89.0 (3.6) | 91.8 (3.6) | 61.0 (1.8) | 71.7 (1.1) | - |
| | ERD | 75.4 (0.9) | 88.8 (0.5) | 71.3 (0.3) | 30.2 (0.5) | - |
| | DCM-Softmax (ours) | 67.3 (0.5) | **86.3 (0.6)** | 74.3 (0.2) | 32.1 (0.8) | 71.8 (0.6) |
| | DCM-MaxLogit (ours) | **66.7 (1.5)** | 87.6 (2.5) | 74.3 (0.5) | 32.2 (1.7) | 71.8 (0.6) |
| | DCM-Energy (ours) | **66.7 (0.5)** | 87.6 (1.1) | 73.9 (0.2) | 32.1 (0.6) | 71.8 (0.6) |

Table 4: Near-OOD detection of ResNet-18 models on the CIFAR-10 and CIFAR-100 datasets. Numbers in parentheses represent the standard error over 5 seeds.

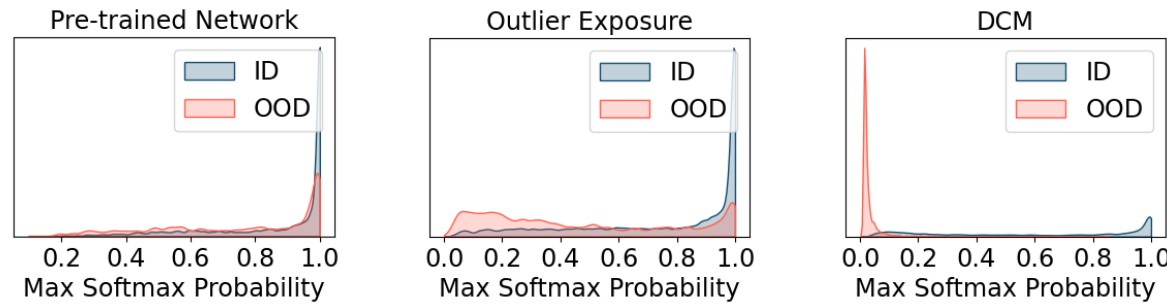

Figure 3: Distribution of maximum softmax probability (left) for ID pre-training, (middle) fine-tuning with OE, (right) fine-tuning with DCM. ID and OOD datasets are CIFAR-100 and TinyImageNet, respectively. DCM results in (1) better separation of predictive confidence for ID and OOD inputs, and (2) low predictive confidence on OOD inputs, suggesting that it learns a conservative model.

Figure 3 suggests that (1) DCM produces a conservative model that is only under-confident on OOD inputs, and (2) DCM better distinguishes ID and OOD inputs by predictive confidence than Outlier Exposure. Further ablations on the robustness of DCM to $\lambda$, the number of finetuning epochs, and the fraction of OOD examples in the uncertainty dataset are in Appendix K.

**DCM is robust to confidence weight.** We fix $\lambda = 0.5$ for all ID-OOD dataset pairs, following prior work (Hendrycks et al., 2018; DeVries & Taylor, 2018; Hendrycks & Gimpel, 2016). While other works (Lee et al., 2017; Liang et al., 2017b) tune hyperparameters for each OOD distribution, we do not in order to test the model's ability to detect OOD inputs from unseen distributions. We plot OOD detection performance for 4 representative ID-OOD dataset pairs with different $\lambda$ in Figure 4 and Figure 5. While $\lambda = 0.5$ is not always optimal, we find that differences in performance due to changes in $\lambda$ are negligible. This suggests that DCM can maintain high performance without extensive hyperparameter tuning, making it more practical for real-world applications.

**Uncertainty dataset composition.** We expect uncertainty datasets with a larger fraction of OOD examples to result in better separation of ID and OOD inputs. Intuitively, minimizing confidence only on OOD inputs should result in the most conservative models. As expected, we observe improved performance with larger proportions of OOD examples in the uncertainty dataset for all methods. We note that DCM achieves the highest performance across all proportions. This holds for several near OOD detection tasks, as illustrated in Figure 4 (left panel) and Figure 6. DCM outperforms binary classifier, and performs similarly to ERD

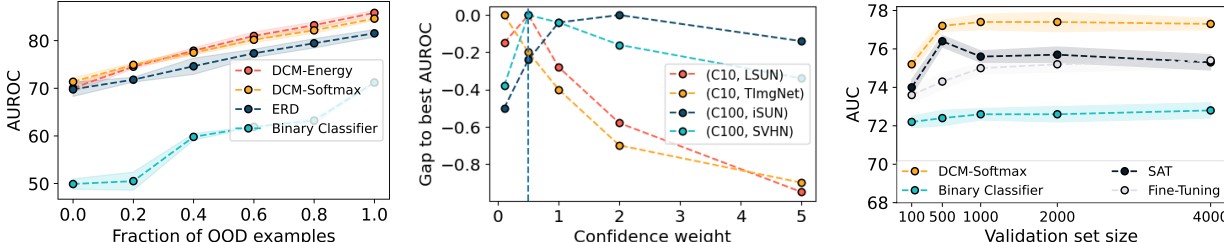

Figure 4: **Robustness of DCM to hyperparameters.** *Left:* Performance of DCM on a near-OOD detection task (CIFAR-100 [0:50] vs CIFAR-100 [50:100]) with various OOD proportions in the uncertainty dataset. Our methods, DCM-Energy and DCM-Softmax, outperform existing methods across all OOD proportions. *Middle:* Relative AUROC of DCM with various confidence weights $\lambda$; note the negligible differences in AUROC. *Right:* Selective classification performance of DCM with uncertainty datasets of various sizes on CIFAR-10 → CIFAR-10-C. These plots suggest that DCM is robust to a range of confidence weights and sizes and compositions in the uncertainty dataset.

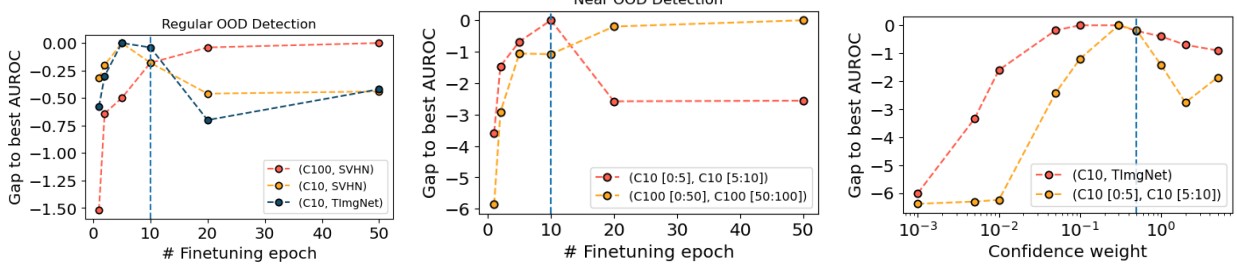

Figure 5: **Further ablations on robustness of DCM to hyperparameters.** *Left:* Relative AUROC of DCM on 3 regular OOD detection setting, where we vary the number of epochs in the second fine-tuning stage. Our default choice of 10 does not generally achieve the best performance. *Middle:* Similar to the plot on the left, but we experiment in the challenging near-OOD detection setting. *Right:* Relative AUROC of DCM where we vary the confidence weight $\lambda$.

while using 1/3 the compute. This suggests that the benefits of data-driven regularization is robust to the uncertainty dataset composition.

**DCM performs competitively in the transductive setting.** We evaluate DCM with the test set as the uncertainty dataset in Table 5 and Table 6. This transductive variant of DCM still performs competitively to prior methods. However, there is a slight drop in performance compared to standard version of DCM due to minimizing confidence on test ID inputs.

# 7 Conclusion

In this work, we propose Data-Driven Confidence Minimization (DCM), which trains models to make conservative predictions by minimizing confidence on an uncertainty dataset.

Our empirical results demonstrate that DCM can lead to more robust classifiers, particularly in conditions of distribution shift. In our experiments, DCM consistently outperformed state-of-the-art methods for selective classification and OOD detection. We believe that the theoretical guarantees and strong empirical performance of DCM represents a promising step towards building more robust and reliable machine learning systems in safety-critical scenarios.

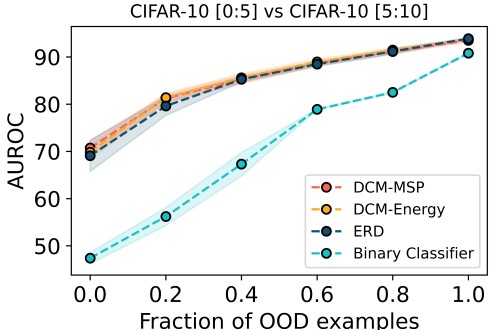

Figure 6: Performance of DCM on CIFAR-10 [0:5] vs CIFAR-10 [5:10] near-OOD detection task with various OOD proportions in the uncertainty dataset. The test dataset is fixed with 2500 ID and 500 OOD examples and is disjoint from the uncertainty dataset.

| Method | Setting | SVHN | | TinyImageNet | | LSUN | | iSUN | |
|---|---|---|---|---|---|---|---|---|---|
| | | AUROC | FPR@95 | AUROC | FPR@95 | AUROC | FPR@95 | AUROC | FPR@95 |
| DCM-Softmax | Regular | 99.5 (0.2) | 1.0 (0.6) | 99.3 (0.1) | 4.1 (1.3) | 99.7 (0.1) | 0.9 (0.3) | 99.5 (0.1) | 1.7 (0.4) |
| | Transductive | 99.8 (0.1) | 0.4 (0.4) | 98.8 (0.3) | 6.5 (2.0) | 99.2 (0.3) | 4.2 (1.5) | 99.2 (0.1) | 4.9 (1.3) |
| DCM-MaxLogit | Regular | 99.5 (0.1) | 0.9 (0.3) | 99.3 (0.1) | 2.7 (0.8) | 99.8 (0.1) | 0.8 (0.4) | 99.4 (0.1) | 1.5 (0.6) |
| | Transductive | 99.9 (0.1) | 0.3 (0.3) | 98.8 (0.2) | 5.9 (1.9) | 99.3 (0.2) | 3.6 (1.3) | 99.2 (0.1) | 4.6 (1.3) |
| DCM-Energy | Regular | 99.5 (0.2) | 0.6 (0.5) | 99.3 (0.1) | 3.2 (1.0) | 99.8 (0.1) | 0.4 (0.1) | 99.5 (0.1) | 1.2 (0.3) |
| | Transductive | 99.9 (0.1) | 0.2 (0.2) | 98.9 (0.2) | 5.2 (1.8) | 99.4 (0.2) | 2.5 (0.8) | 99.3 (0.1) | 3.5 (0.5) |

Table 5: Comparison between the regular and transductive setting peformance of our method for ResNet-18 models trained on CIFAR-10.

| Method | Setting | SVHN | | TinyImageNet | | LSUN | | iSUN | |
|---|---|---|---|---|---|---|---|---|---|
| | | AUROC (↑) | FPR@95 (↓) | AUROC (↑) | FPR@95 (↓) | AUROC (↑) | FPR@95 (↓) | AUROC (↑) | FPR@95 (↓) |
| DCM-Softmax | Regular | 99.3 (0.1) | 2.5 (1.4) | 98.7 (0.3) | 7.8 (2.5) | 99.4 (0.2) | 1.6 (1.5) | 98.8 (0.4) | 6.3 (3.3) |
| | Transductive | 99.3 (0.4) | 2.8 (2.5) | 97.6 (0.4) | 16.3 (4.6) | 98.3 (0.9) | 9.3 (7.3) | 97.9 (1.2) | 12.0 (8.2) |
| DCM-MaxLogit | Regular | 99.2 (0.1) | 3.7 (0.6) | 98.8 (0.2) | 6.4 (1.8) | 99.6 (0.1) | 0.7 (0.3) | 99.2 (0.2) | 3.1 (1.7) |
| | Transductive | 99.5 (0.4) | 1.6 (1.8) | 97.7 (0.3) | 14.8 (4.4) | 98.5 (0.8) | 7.7 (5.9) | 98.2 (0.9) | 10.2 (6.3) |
| DCM-Energy | Regular | 99.5 (0.1) | 1.0 (0.6) | 99.0 (0.3) | 4.9 (2.3) | 99.6 (0.2) | 0.8 (0.7) | 99.2 (0.3) | 2.4 (0.9) |
| | Transductive | 99.5 (0.3) | 1.4 (1.6) | 97.8 (0.3) | 11.9 (5.1) | 98.7 (0.5) | 5.3 (3.5) | 98.7 (0.4) | 6.1 (3.5) |

Table 6: Comparison between the regular and transductive setting peformance of our method for ResNet-18 models trained on CIFAR-100.

The requirement of an uncertainty dataset that covers regions that the model may mis-classify can preclude some applications. Furthermore, the theoretical guarantees for DCM only apply to inputs that are represented by the uncertainty dataset and DCM requires separate fine-tuning for each different test distribution. Future work can develop better methods for gathering or constructing uncertainty datasets to make the framework more widely applicable and increase performance. It would also be interesting to extend DCM to the problem of selective classification with OOD detection (Jaeger et al., 2022; Xia & Bouganis, 2022).

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

## A    Theoretical Analysis

In this section we provide a simple theoretical setup for our algorithm. First, we show our algorithm can perfectly detect unknown examples when the known examples in the test set also appears in this training set. Next, we show that under the assumptions of function smoothness and closeness of known train and test examples in the input space, this also holds for unseen known and unknown examples.

### A.1    Problem Setting

Let $\mathcal{X}$ be the input space and $\mathcal{Y}$ the label space. Let $\mathcal{P}_{\mathrm{ID}}$ be a distribution over $\mathcal{X} \times \{1, \ldots, C\} \subseteq \mathcal{X} \times \mathcal{Y}$ i.e., there are $C$ classes, and let $D_{\mathrm{tr}}$ be a training dataset consisting of $n$ datapoints sampled from $\mathcal{P}_{\mathrm{ID}}$. We train a classifier $f_\theta : \mathcal{X} \to [0,1]^C$ on the training data. We also consider a different distribution $\mathcal{P}_{\mathrm{OOD}}$ over $\mathcal{X} \times \mathcal{Y}$ that is different from $\mathcal{P}_{\mathrm{ID}}$ (the OOD distribution). Let $D_{\mathrm{u}}$ be an unlabeled test set where half the examples are sampled from $\mathcal{P}_{\mathrm{ID}}$, the other half are sampled from $\mathcal{P}_{\mathrm{OOD}}$. Our objective is to minimize the following loss function:

$$\mathcal{L}(\theta) = \underset{(x,y) \in D_{\mathrm{tr}}}{\mathbb{E}} \left[ \mathcal{L}_{\mathrm{xent}}(f_\theta(x), y) \right] + \lambda \underset{x' \in D_{\mathrm{u}}}{\mathbb{E}} \left[ \mathcal{L}_{\mathrm{con}}(f_\theta(x')) \right], \tag{5}$$

where $\lambda > 0$, $\mathcal{L}_{\mathrm{xent}}$ is the standard cross-entropy loss, and $\mathcal{L}_{\mathrm{con}}$ is a confidence loss which is calculated as the cross-entropy with respect to the uniform distribution over the $C$ classes. We focus on the maximum softmax probability $\mathrm{MSP}(p) \triangleq \max_i p_i$ as a measure of confidence in a given categorical distribution $p$.

### A.2    Simplified Setting: known Examples Shared Between Train and Unlabeled Sets

We start with the following lemma which characterizes the interaction of our loss function (3) with a single datapoint.

**Proposition A.1** (Lower bound on true confidence). *Let $p$ be the true label distribution of input $x$. The minimum of the objective function (3) is achieved when the predicted distribution is $p_\lambda \triangleq \frac{p + \lambda \frac{1}{C}}{1 + \lambda}$. For all $x$ within $D_u$ and $D_{tr}$, the optimal distribution $p_\lambda$ satisfies $MSP(p_\lambda) \leq MSP(p)$, with equality iff $\lambda = 0$.*

*Proof.* Denote the predicted logits for input $x$ as $z \in \mathbb{R}^C$, and softmax probabilities as $s = e^z / \sum_i e^{z_i} \in [0,1]^C$. The derivative of the logits with respect to the two loss terms have the closed-form expressions $\frac{\partial}{\partial z} \mathcal{L}_{\mathrm{xent}} = s - p$, $\frac{\partial}{\partial z} \mathcal{L}_{\mathrm{con}} = s - \frac{1}{C} \mathbf{1}$. Setting the derivative of the overall objective to zero, we have

$$\frac{\partial}{\partial z} \left( \mathcal{L}_{\mathrm{xent}} + \lambda \mathcal{L}_{\mathrm{con}} \right) = s - p + \lambda \left( s - \frac{\mathbf{1}}{C} \right) = 0 \implies s = \frac{p + \lambda \frac{\mathbf{1}}{C}}{1 + \lambda} = p_\lambda. \tag{6}$$

To check the lower bound property, note that $p_\lambda$ is a combination of $p$ and the uniform distribution $U$, where $U$ is the uniform distribution over the $C$ classes and has the lowest possible MSP among all categorical distributions over $C$ classes. $\square$

The resulting predictive distribution $p_\lambda$ can alternatively be seen as Laplace smoothing with pseudocount $\lambda$ applied to the true label distribution $p$. This new distribution can be seen as "conservative" in that it (1) has lower MSP than that of $p$, and (2) has an entropy greater than that of $p$.

**Lemma A.2** (Pinsker's inequality). *If $P$ and $Q$ are two probability distributions, then*

$$\delta_{TV}(P, Q) \leq \sqrt{\frac{1}{2} D_{KL}(P \parallel Q)}, \tag{7}$$

*where $\delta_{TV}(P, Q)$ is the total variation distance between $P$ and $Q$.*

*Proof.* Refer to (Pinsker, 1964; Canonne, 2022) for a detailed proof. $\square$

**Lemma A.3** (Low loss implies separation, transductive case). *Assume that all known examples in $D_u$ are also in $D_{tr}$, and that $\mathcal{D}_{in} \cap \mathcal{D}_{out} = \varnothing$. Let $D_{in}^{test} = \{x \in D^{test} : x \sim \mathcal{D}_{in}\}(= D^{train})$ and $D_{out}^{test} = \{x \in D^{test} : x \sim \mathcal{D}_{out}\} = D^{test} \backslash D^{train}$. Let $\mathcal{L}_0$ be the lowest achievable loss for the objective (3) with $\lambda > 0$. Then there exists $\epsilon > 0$ such that $\mathcal{L}(\theta) - \mathcal{L}_0 < \epsilon$ implies the following relationship between the max probabilities holds:*

$$\min_{x \in D_{in}^{test}} MSP(f_\theta(x)) > \max_{x \in D_{out}^{test}} MSP(f_\theta(x)) \tag{8}$$

*Proof.* Since the training set is a subset of the unlabeled set, we can rearrange the objective (3) as

$$\mathcal{L}(\theta) = \mathbb{E}_{(x,y) \in D_{in}^{test}} \left[ \mathcal{L}_{\text{xent}}(f_\theta(x), y) + \lambda \mathcal{L}_{\text{con}}(f_\theta(x)) \right] + \mathbb{E}_{x \in D_{out}^{test}} \left[ \lambda \mathcal{L}_{\text{con}}(f_\theta(x')) \right]. \tag{9}$$

Note that the first term is the cross-entropy between $f_\theta(x)$ and $p_\lambda \triangleq \frac{p + \lambda \frac{1}{C}}{1 + \lambda}$, and the second term is the cross-entropy between $f_\theta(x)$ and the uniform distribution $U$. We now rearrange to see that

$$\mathcal{L}(\theta) - \mathcal{L}_0 = \mathbb{E}_{(x,y) \in D_{in}^{test}} \left[ D_{KL}(p_\lambda \parallel f_\theta(x)) \right] + \mathbb{E}_{x \in D_{out}^{test}} \left[ D_{KL}(U \parallel f_\theta(x)) \right], \tag{10}$$

where the lowest achievable loss $\mathcal{L}_0$ is obtained by setting $f_\theta(x) = p_\lambda$ for known inputs and $f_\theta(x) = U$ for unknown inputs. Because $\mathcal{L} - \mathcal{L}_0 < \epsilon$, we know that $D_{KL}(p_\lambda \parallel f_\theta(x)) < N\epsilon$ for all known inputs and $D_{KL}(U \parallel f_\theta(x)) < N\epsilon$ for all unknown inputs.

By Lemma A.2, we have for known input $x$

$$\delta_{\text{TV}}(p_\lambda, f_\theta(x)) \leq \sqrt{\frac{1}{2} D_{KL}(p_\lambda \parallel f_\theta(x))} = \sqrt{\frac{N\epsilon}{2}}. \tag{11}$$

By the triangle inequality and because MSP is 1-Lipschitz with respect to output probabilities, we have for all known inputs

$$\text{MSP}(f_\theta(x)) \geq \text{MSP}(p_\lambda) - \sqrt{\frac{N\epsilon}{2}} = \frac{1}{1 + \lambda} + \frac{\lambda}{1 + \lambda} \frac{1}{C} - \sqrt{\frac{N\epsilon}{2}}. \tag{12}$$

Similarly, by Lemma A.2, we have for unknown input $x$

$$\delta_{\text{TV}}(U, f_\theta(x)) \leq \sqrt{\frac{1}{2} D_{KL}(U \parallel f_\theta(x))} = \sqrt{\frac{N\epsilon}{2}}. \tag{13}$$

By the triangle inequality and because MSP is 1-Lipschitz with respect to output probabilities, we have for all unknown inputs

$$\text{MSP}(f_\theta(x)) \leq \text{MSP}(U) + \sqrt{\frac{N\epsilon}{2}} = \frac{1}{C} + \sqrt{\frac{N\epsilon}{2}}. \tag{14}$$

Letting $\epsilon < \frac{1}{2N} \left( \frac{C-1}{(1+\lambda)C} \right)^2$, we have

$$\min_{x \in D_{in}^{test}} \text{MSP}(f_\theta(x)) \geq \frac{1}{1 + \lambda} + \frac{\lambda}{1 + \lambda} \frac{1}{C} - \sqrt{\frac{N\epsilon}{2}} > \frac{1}{C} + \sqrt{\frac{N\epsilon}{2}} \geq \max_{x \in D_{out}^{test}} \text{MSP}(f_\theta(x)). \tag{15}$$

$\square$

Lemma A.3 shows that in the transductive setting, minimizing our objective $L(\theta)$ (3) below some threshold provably leads to a separation between known and unknown examples in terms of the maximum predicted probability for each example.

### A.3 More general setting

We prove a more general version of the claim in Lemma A.3 which applies to datapoints outside of the given dataset $D^{test}$. Our theorem below depends only on a mild smoothness assumption on the learned function.

**Proposition A.4** (Low loss implies separation). *Assume that all known examples in $D_u$ are also in $D_{tr}$, and that $\mathcal{D}_{in} \cap \mathcal{D}_{out} = \varnothing$. Let $D_{in}^{test} = \{x \in D^{test} : x \sim \mathcal{D}_{in}\}(= D^{train})$ and $D_{out}^{test} = \{x \in D^{test} : x \sim \mathcal{D}_{out}\} = D^{test} \backslash D^{train}$. Assume that the classifier $f_\theta : \mathcal{X} \to [0,1]^C$ is $K$-Lipschitz continuous for all $\theta$, i.e., for all $x, x' \in \mathcal{X}$, $||f_\theta(x) - f_\theta(x')||_\infty \leq Kd(x,x')$ for some constant $K > 0$. Let $\mathcal{L}_0$ be the lowest achievable loss for the objective (3) with $\lambda > 0$. For $\delta > 0$, denote the union of $\delta$-balls around the known and unknown datapoints as*

$$D_{in}^\delta \triangleq \{x | \exists x' \in D_{in}^{test} \text{ s.t. } d(x,x') < \delta\}, \quad D_{out}^\delta \triangleq \{x | \exists x' \in D_{out}^{test} \text{ s.t. } d(x,x') < \delta\}. \tag{16}$$

*Then there exists $\epsilon, \delta > 0$ such that $\mathcal{L}(\theta) - \mathcal{L}_0 < \epsilon$ implies the following relationship between the max probabilities holds:*

$$\inf_{x \in D_{in}^\delta} MSP(f_\theta(x)) > \sup_{x \in D_{out}^\delta} MSP(f_\theta(x)) \tag{17}$$

*Proof.* By Lemma A.3, we have for some $\epsilon$, $\min_{x \in D_{in}^{test}} \text{MSP}(f_\theta(x)) > \max_{x \in D_{out}^{test}} \text{MSP}(f_\theta(x))$. Fix $\epsilon$ and denote the difference of these two terms as

$$\min_{x \in D_{in}^{test}} \text{MSP}(f_\theta(x)) - \max_{x \in D_{out}^{test}} \text{MSP}(f_\theta(x)) = \Delta. \tag{18}$$

For any $x_{\text{in}}^\delta \in D_{in}^\delta$ and $x_{\text{out}}^\delta \in D_{out}^\delta$, let $x_{\text{in}} \in D_{in}^{test}, x_{\text{out}} \in D_{out}^{test}$ satisfy $d(x_{\text{in}}^\delta, x_{\text{in}}) < \delta$ and $d(x_{\text{out}}^\delta, x_{\text{out}}) < \delta$. By the $K$-Lipschitz property, we have

$$\text{MSP}(f_\theta(x_{\text{in}}^\delta)) \geq \text{MSP}(f_\theta(x_{\text{in}})) - K\delta, \quad \text{MSP}(f_\theta(x_{\text{out}}^\delta)) \leq \text{MSP}(f_\theta(x_{\text{out}})) + K\delta. \tag{19}$$

Setting $\delta < \frac{\Delta}{2K}$, we have

$$\text{MSP}(f_\theta(x_{\text{in}}^\delta)) \geq \text{MSP}(f_\theta(x_{\text{in}})) - K\delta > \text{MSP}(f_\theta(x_{\text{out}})) + K\delta \geq \text{MSP}(f_\theta(x_{\text{out}}^\delta)). \tag{20}$$

Since the choice of $x_{\text{in}}^\delta$ and $x_{\text{out}}^\delta$ was arbitrary, the equation above holds for all datapoints inside each $\delta$-ball. Therefore, we have

$$\inf_{x \in D_{in}^\delta} \text{MSP}(f_\theta(x)) > \sup_{x \in D_{out}^\delta} \text{MSP}(f_\theta(x)). \tag{21}$$

$\square$

## B Metrics

### B.1 OOD Detection

We first define precision, recall, true positive rate and false positive rate. Let TP and FP denote the number of examples correctly and incorrectly classified as positive, respectively. Similarly, let TN and FN denote the number of examples correctly and incorrectly classified as negative, respectively.

*Precision* is defined as the fraction of correctly classified positive examples, among all examples that are classified as positive.

$$\text{Precision} = \frac{\text{TP}}{\text{TP} + \text{FP}}$$

*Recall* (also referred to as *true positive rate (TPR)*) is defined as the fraction of correctly classified positive examples among all positive examples.

$$\text{Recall} = \frac{\text{TP}}{\text{TP} + \text{FN}}$$

*False positive rate (FPR)* is defined as

$$\text{FPR} = \frac{\text{FP}}{\text{FP} + \text{TN}}.$$

We use the following metrics to evaluate OOD detection performance.

1. **AUROC**: The receiver operating characteristic (ROC) curve is obtained by plotting the true positive rate vs the false positive rate at different thresholds. AUROC is the area under the ROC curve. AUROC is always between 0 and 1; the AUROC of a random and a perfect classifier is 0.5 and 1.0 respectively. The higher the AUROC, the better.

2. **AUPR**: The precision-recall (PR) curve is obtained by plotting the precision and recall of a classifier at different threshold settings. AUPR is the area under this PR curve. Similar to AUROC, higher AUPR implies a better classifier. See that AUPR would be different based on whether we label the ID examples as positive or vice-versa. In this context, AUPR-In and AUPR-Out refers to AUPR calculated using the convention of denoting ID and OOD examples as positive respectively. If not mentioned otherwise, AUPR in this paper refers to AUPR-Out.

3. **FPR@TPR**: This metric represents the false positive rate of the classifier, when the decision threshold is chosen such that true positive rate is TPR%. Typically, we report FPR@95 in our paper, following prior work such as Hendrycks et al. (2022).

### B.2 Selective Classification

1. **ECE**: The expected calibration error (ECE) measures the calibration of the classifier. It is calculated as the expected difference between confidence and accuracy, i.e., $\mathbb{E}[|p(\hat{y} = y \mid \hat{p} = p) - p|]$.

2. **Acc@Cov**: This metric measures the average accuracy of a fixed fraction of most confident datapoints. Specifically, we calculate the average accuracy on the Cov% datapoints with highest confidence.

3. **Cov@Acc**: This metric measures size of the largest subset that achieves a given average accuracy. Specifically, we calculate the largest fraction of data for which selective accuracy is above Acc.

4. **AUC**: The area under the curve of selective classification accuracy vs coverage.

## C   Variants of DCM for OOD Detection

For OOD detection, we experiment with three different scoring methods on top of DCM. Concretely, we denote the input space as $\mathcal{X}$ and assume that our ID distribution has $C$ classes. Further, let $f : \mathcal{X} \to \mathbb{R}^C$ represent our model, and $S : \mathcal{X} \to \mathbb{R}$ represent a score function. Then OOD detection becomes a binary classification problem, where we use the convention that OOD examples are positive and ID examples are negative. During test time, we would choose a threshold $\gamma$ and for $x \in \mathcal{X}$, we say $x$ is OOD if $S(x) \geq \gamma$, and $x$ is classified as ID otherwise. We experiment with three commonly used choices for the score function, $S$.

1. **Maximum softmax score (MSP)** (Hendrycks & Gimpel, 2016; Vaze et al., 2022): For class $i \in \{1, \ldots, C\}$, the softmax score, $S^i_{\text{soft}}(x)$ is defined as:

$$S^i_{\text{soft}}(x) = \frac{\exp\left(f^i(x)\right)}{\sum_{j=1}^{C} \exp\left(f^j(x)\right)}$$

The MSP score is defined as:

$$S_{\text{MSP}}(x) = - \max_{i \in \{1, \ldots, C\}} S^i_{\text{soft}}(x)$$

Here the negative signs comes due to our convention of labeling OOD examples as positive.

2. **MaxLogit (Hendrycks et al., 2022; Vaze et al., 2022)**: Instead of using the softmax probabilities, we use the maximum of the model's un-normalized outputs (logits) as the score. Formally,

$$S_{\text{maxlogit}}(x) = - \max_{i \in 1,...,C} f^i(x)$$

3. **Energy (Liu et al., 2020)**: The energy score is defined as follows:

$$S_{\text{energy}}(x) = - \log \left( \sum_{i=1}^{C} e^{f^i(x)} \right)$$

We see in our experiments that all three scores, when combined with DCM framework, performs similarly, with Energy score giving slightly better performance.

## D   Detailed OOD Detection Results in the Regular Setting

### D.1   Baselines

We compare DCM against several prior OOD detection methods.

- **MSP** (Hendrycks & Gimpel, 2016; Vaze et al., 2022): A simple baseline for OOD detection, where we take a network trained on ID samples and threshold on the network's maximum softmax probability prediction on a test example to separate ID and OOD examples.

- **Max Logit** (Hendrycks et al., 2022; Vaze et al., 2022): Similar to MSP, but instead of using normalized softmax probabilities, this uses the maximum of the output logits to perform OOD detection.

- **ODIN** (Liang et al., 2017a): This method uses temperature scaling and adding small noise perturbations to the inputs to increase the separation of softmax probability between ID and OOD examples.

- **Mahalanobis** (Lee et al., 2018): This method takes a pretrained softmax classifier and uses the mahalanobis distance in the embedding space to separate ID examples from OOD examples.

- **Energy Score** (Liu et al., 2020): Instead of the softmax probability, this method uses energy scores to separate ID and OOD examples.

- **Outlier Exposure** (Hendrycks et al., 2018): Leverages examples from a pseudo-OOD distribution, i.e., a distribution different from the training distribution but not necessarily the OOD distribution seen at test-time. Fine-tunes a pre-trained network with a combined objective of (1) cross entropy loss on the training examples, and (2) confidence minimization loss on the pseudo-OOD examples.

- **Energy Based Fine-Tuning** (Liu et al., 2020): Minimizes the energy-based confidence score on pseudo-OOD examples.

- **WOODS** (Katz-Samuels et al., 2022): Leverages a "wild" dataset – naturally comprising both in-distribution (ID) and OOD samples. Rather than using confidence minimization, WOODS formulates a constrained optimization problem to maximize the OOD detection rate while constraining classification error for ID data and OOD error rate for ID examples.

  We note that our experimental setup differs from that of WOODS. We closely follow the setup established by (Hendrycks et al., 2018). First, unlike DCM, the baseline approach in Katz-Samuels et al. (2022) does not use random augmentations on the unlabeled set to prevent overfitting. Second, at each gradient update, WOODS computes the combined objective using a 10:1 ratio of ID:uncertainty set, whereas we use a 1:1 sampling ratio. Third, WOODS uses a mixed ID-OOD validation set for hyperparameter tuning and early stopping, while DCM and Outlier Exposure (Hendrycks et al., 2018) only use an ID validation set.

### D.2    ID Datasets

We use the following ID datasets from common benchmarks:

- **CIFAR-10** (Krizhevsky et al., a): CIFAR-10 contains 50,000 train and 10,000 test images, separated into 10 disjoint classes. The images have 3 channels and are of size 32 x 32. The classes are similar but disjoint from CIFAR-100.

- **CIFAR-100** (Krizhevsky et al., b): Similar to CIFAR-10 and contains 50,000 train and 10,000 test images. However, the images are now separated into 100 fine-grained and 20 coarse (super) classes. Each super-class contains 5 fine-grained classes.

### D.3    OOD Datasets

In addition to CIFAR-10 and CIFAR-100, we follow prior work (Tajwar et al., 2021; Hendrycks & Gimpel, 2016; Liu et al., 2020) and use the following benchmark OOD detection dataset:

- **SVHN** (Netzer et al., 2011): SVHN contains images of the 10 digits in English which represent the 10 classes in the dataset. The dataset contains 73,257 train and 26,032 test images. The original dataset also contains extra training images that we do not use for our experiments. Each image in the dataset has 3 channels and has shape 32 x 32.

- **TinyImageNet (resized)** (Le & Yang, 2015; Deng et al., 2009; Liang et al., 2017a): TinyImageNet contains 10,000 test images divided into 200 classes and is a subset of the larger ImageNet (Deng et al., 2009) dataset. The original dataset contains images of shape 64 x 64 and Liang et al. (2017a) further creates a dataset by randomly cropping and resizing the images to shape 32 x 32. We use the resized dataset here for our experiments.

- **LSUN (resized)** (Yu et al., 2015; Liang et al., 2017a): The **L**arge-scale **S**cene **UN**derstanding dataset (LSUN) contains 10,000 test images divided into 10 classes. Similar to the TinyImageNet dataset above, Liang et al. (2017a) creates a dataset by randomly cropping and resizing the images to shape 32 x 32. We use the resized dataset here for our experiments.

- **iSUN** (Xu et al., 2015; Liang et al., 2017a): iSUN contains 6,000 training, 926 validation and 2,000 test images. We use the same dataset used by Liang et al. (2017a).

Instructions on how to download the TinyImageNet, LSUN and iSUN datasets can be found here: `https://github.com/ShiyuLiang/odin-pytorch`

### D.4    Training Details

- **Architecture**: For all experiments in this section, we use a WideResNet-40-2 (Zagoruyko & Komodakis, 2016) network.

- **Hyper-parameters**: Outlier exposure and energy based fine-tuning uses 80 Million Tiny Images (Torralba et al., 2008) as the pseudo-OOD dataset. This dataset has been withdrawn because it contains derogatory terms as categories. Thus, for fair comparison, we use the pre-trained weights provided by these papers' authors for our experiments. For MSP, ODIN, Mahalanobis and energy score, we train our networks for 110 epochs with an initial learning rate of 0.1, weight decay of $5 \times 10^{-4}$, dropout 0.3 and batch size 128. ODIN and Mahalanobis require a small OOD validation set to tune hyper-parameters. Instead, we tune the hyper-parameters over the entire test set and report the best numbers, since we only want an upper bound on the performance of these methods. For ODIN, we try $T \in \{1, 10, 100, 1000\}$ and $\epsilon \in \{0.0, 0.0005, 0.001, 0.0014, 0.002, 0.0024, 0.005, 0.01, 0.05, 0.1, 0.2\}$ as our hyper-parameter search grid, and for Mahalanobis, we use the same hyper-parameter grid for $\epsilon$. For the WOODS baseline, we use all default hyperparameters, except that our setting uses an unlabeled auxiliary set with OOD proportion

| ID Dataset / Network | Method | SVHN | | TinyImageNet | | LSUN | | iSUN | |
|---|---|---|---|---|---|---|---|---|---|
| | | AUROC (↑) | FPR@95 (↓) | AUROC (↑) | FPR@95 (↓) | AUROC (↑) | FPR@95 (↓) | AUROC (↑) | FPR@95 (↓) |
| CIFAR-10 WRN-40-2 | MSP | 87.2 (5.6) | 43.4 (23.3) | 90.3 (1.4) | 32.8 (6.0) | 93.3 (0.9) | 21.3 (2.6) | 92.0 (1.3) | 25.9 (4.1) |
| | MaxLogit | 88.5 (2.4) | 42.1 (9.4) | 93.2 (1.2) | 27.3 (4.9) | 96.6 (0.8) | 14.1 (2.7) | 95.5 (0.8) | 18.4 (2.7) |
| | ODIN | 90.3 (2.1) | 41.1 (8.1) | 93.8 (0.7) | 27.6 (6.5) | 97.5 (0.9) | 10.9 (3.6) | 96.6 (0.7) | 16.8 (2.7) |
| | Mahalanobis | 97.3 (0.7) | 14.7 (4.4) | 91.2 (1.3) | 38.9 (4.6) | 92.1 (0.6) | 28.6 (0.8) | 93.7 (1.4) | 26.8 (5.4) |
| | Energy Score | 82.8 (10.5) | 59.7 (22.7) | 92.0 (2.8) | 34.0 (10.9) | 96.2 (1.2) | 16.0 (5.0) | 94.9 (1.9) | 23.2 (8.6) |
| | VOS | 90.8 (1.4) | 28.4 (7.3) | 93.4 (0.7) | 27.3 (2.7) | 97.0 (0.3) | 12.8 (1.6) | 96.0 (0.6) | 16.4 (2.3) |
| | WOODS | 99.5 (0.0) | 3.3 (0.3) | 99.2 (0.1) | 5.3 (0.6) | 99.3 (0.1) | 5.0 (0.5) | 99.5 (0.1) | 2.9 (0.4) |
| | Outlier Exposure | 98.5 | 4.8 | 97.4 | 13.0 | 99.1 | 3.7 | 99.1 | 5.0 |
| | Energy Fine-Tuning | 99.3 | 2.1 | 98.2 | 7.0 | 99.3 | 1.9 | 99.4 | 2.6 |
| | DCM-Softmax (ours) | **99.7 (0.1)** | 0.4 (0.3) | **99.3 (0.3)** | 2.6 (1.6) | **99.8 (0.1)** | 0.5 (0.4) | **99.7 (0.1)** | 0.6 (0.2) |
| | DCM-MaxLogit (ours) | **99.8 (0.1)** | 0.3 (0.1) | **99.5 (0.2)** | 1.9 (0.6) | **99.9 (0.1)** | **0.2 (0.1)** | **99.8 (0.1)** | 0.5 (0.2) |
| | DCM-Energy (ours) | **99.8 (0.1)** | **0.1 (0.1)** | **99.4 (0.2)** | **1.0 (0.8)** | **99.9 (0.1)** | **0.1 (0.1)** | **99.8 (0.1)** | **0.1 (0.1)** |
| CIFAR-10 ResNet-18 | Binary Classifier | 98.9 (0.2) | 1.3 (1.0) | 98.7 (0.6) | **1.8 (3.8)** | 99.0 (0.3) | **0.3 (0.6)** | 98.8 (0.8) | 1.6 (2.5) |
| | ERD | 99.3 (0.2) | 1.7 (1.2) | **99.3 (0.1)** | **1.7 (0.6)** | **99.7 (0.1)** | 0.2 (0.2) | **99.7 (0.2)** | **0.5 (0.4)** |
| | DCM-Softmax (ours) | **99.5 (0.2)** | 1.0 (0.6) | **99.3 (0.1)** | 4.1 (1.3) | **99.7 (0.1)** | 0.9 (0.3) | **99.5 (0.1)** | 1.7 (0.4) |
| | DCM-MaxLogit (ours) | **99.5 (0.1)** | 0.9 (0.3) | **99.3 (0.1)** | 2.7 (0.8) | **99.8 (0.1)** | 0.8 (0.4) | 99.4 (0.1) | 1.5 (0.6) |
| | DCM-Energy (ours) | **99.5 (0.2)** | **0.6 (0.5)** | **99.3 (0.1)** | 3.2 (1.0) | **99.8 (0.1)** | 0.4 (0.1) | **99.5 (0.1)** | 1.2 (0.3) |
| CIFAR-100 WRN-40-2 | MSP | 77.7 (1.4) | 58.0 (4.9) | 68.0 (3.2) | 77.0 (5.7) | 68.5 (1.5) | 75.6 (3.7) | 67.1 (2.4) | 77.6 (3.7) |
| | MaxLogit | 84.3 (2.8) | 43.2 (7.4) | 74.7 (5.5) | 72.7 (11.3) | 75.9 (4.8) | 67.3 (10.4) | 75.4 (4.5) | 70.9 (10.0) |
| | ODIN | 90.9 (2.0) | 30.7 (3.7) | 82.2 (4.9) | 62.3 (11.7) | 82.9 (5.0) | 57.1 (11.3) | 82.0 (2.6) | 61.7 (8.1) |
| | Mahalanobis | 92.7 (1.2) | 32.3 (6.0) | 91.8 (2.0) | 39.0 (7.2) | 92.2 (2.3) | 34.3 (10.0) | 89.6 (3.8) | 43.5 (10.2) |
| | Energy Score | 81.7 (2.4) | 51.3 (5.8) | 73.1 (4.3) | 73.5 (10.5) | 75.2 (4.9) | 70.0 (8.7) | 73.8 (3.8) | 72.0 (8.2) |
| | VOS | 85.1 (1.3) | 41.0 (1.9) | 78.2 (2.7) | 63.3 (4.9) | 80.1 (2.2) | 56.7 (7.0) | 79.1 (2.9) | 59.2 (7.3) |
| | WOODS | 98.6 (0.0) | 8.6 (0.3) | 97.5 (0.0) | 18.3 (0.3) | 98.4 (0.2) | 8.7 (1.3) | 97.5 (0.3) | 16.0 (2.1) |
| | Outlier Exposure | 88.2 | 40.4 | 75.7 | 71.6 | 81.4 | 59.1 | 79.2 | 66.4 |
| | Energy Fine-Tuning | 96.8 | 12.6 | 70.9 | 85.2 | 80.9 | 65.6 | 77.4 | 75.1 |
| | DCM-Softmax (ours) | **99.6 (0.1)** | **0.6 (0.7)** | 98.7 (0.3) | **5.9 (2.9)** | 99.5 (0.2) | 1.1 (1.0) | 99.1 (0.2) | 2.7 (1.9) |
| | DCM-MaxLogit (ours) | **99.6 (0.2)** | 0.8 (1.0) | **99.0 (0.2)** | 3.6 (2.9) | **99.8 (0.1)** | **0.1 (0.1)** | **99.2 (0.3)** | 2.2 (2.6) |
| | DCM-Energy (ours) | **99.7 (0.1)** | **0.3 (0.3)** | **99.0 (0.2)** | 3.5 (2.5) | **99.7 (0.1)** | 0.5 (0.6) | **99.4 (0.2)** | **0.9 (0.6)** |
| CIFAR-100 ResNet-18 | Binary Classifier | 95.1 (6.8) | 25.8 (40.8) | **99.0 (0.7)** | **0.7 (0.6)** | 99.2 (0.4) | **0.0 (0.1)** | 98.3 (0.5) | 4.0 (5.3) |
| | ERD | 99.0 (0.1) | 2.3 (0.9) | 98.8 (0.3) | 5.4 (1.9) | **99.5 (0.1)** | 0.8 (0.4) | **99.2 (0.1)** | **1.7 (0.9)** |
| | DCM-Softmax (ours) | 99.3 (0.1) | 2.5 (1.4) | **98.7 (0.3)** | 7.8 (2.5) | 99.4 (0.2) | 1.6 (1.5) | 98.8 (0.4) | 6.3 (3.3) |
| | DCM-MaxLogit (ours) | 99.2 (0.1) | 3.7 (0.6) | **98.8 (0.2)** | 6.4 (1.8) | **99.6 (0.1)** | 0.7 (0.3) | **99.2 (0.2)** | 3.1 (1.7) |
| | DCM-Energy (ours) | **99.5 (0.1)** | **1.0 (0.6)** | **99.0 (0.3)** | 4.9 (2.3) | **99.6 (0.2)** | 0.8 (0.7) | **99.2 (0.3)** | **2.4 (0.9)** |

Table 7: OOD detection performance of models trained on CIFAR-10 or CIFAR-100 and evaluated on four different OOD datasets. We average metrics across 5 random seeds and show standard error in parentheses. Pre-trained weights provided by the respective authors are used to reproduce outlier exposure and energy fine-tuning results, and hence those results do not have associated standard errors. This is done due to these methods using 80-million tiny images as their auxiliary dataset, which has since been withdrawn and hence these methods' performance cannot be reproduced for other random seeds.

$\pi = 0.2$. We train with a learning rate of 0.001 and batch size of 128 for 100 epochs. We use an in-distribution penalty of 1.0, out-of-distribution penalty of 1.0, classification penalty of 1.0, false alarm cutoff of 0.05, learning rate for updating lambda of 1.0, tolerance for the loss constraint of 2.0, multiplicative factor of 1.5 for the penalty method, and constraint tolerance of 0.0. For our method, we pre-train our network for 100 epochs with the same setup, and fine-tune the network with our modified loss objective for 10 epochs using the same setting, except we use a initial learning rate of 0.001, batch size 32 for ID train set and 64 for the uncertainty dataset. During fine-tuning, we use 27,000 images per epoch, 9,000 of which are labeled ID train examples and the rest are from the uncertainty dataset. Finally, we use $\lambda = 0.5$ for all experiments, as in Hendrycks et al. (2018), without any additional hyper-parameter tuning.

- **Dataset train/val split**: For all methods except outlier exposure and energy based fine-tuning, we use 40,000 out of the 50,000 train examples for training and 10,000 train examples for validation. Note that outlier exposure and energy based fine-tuning uses weights pre-trained with all 50,000 training examples, which puts our method in disadvantage.

- **Uncertainty and test dataset construction**: For our method, we use two disjoint sets of 6,000 images as the uncertainty dataset and test set. Each set contains 5,000 ID examples and 1,000 OOD examples.

- **Augmentations**: For all methods, we use the same standard random flip and random crop augmentations during training/fine-tuning.

| Setting | Method | FPR95 ↓ | FPR99 ↓ | AUROC ↑ | AUPR-In ↑ | AUPR-Out ↑ |
|---|---|---|---|---|---|---|
| ID = CIFAR-100 OOD = CIFAR-10 | MSP | 64.4 (1.4) | 80.5 (0.7) | 74.6 (0.7) | 93.9 (0.2) | 32.8 (1.7) |
| | ODIN | 69.2 (2.4) | 86.6 (4.0) | 75.5 (0.9) | 93.7 (0.4) | 34.6 (0.9) |
| | Mahalanobis | 87.7 (4.2) | 96.7 (0.6) | 59.1 (6.1) | 87.8 (2.4) | 20.4 (2.6) |
| | Energy Score | 67.2 (3.2) | 86.6 (1.6) | 75.7 (0.9) | 93.8 (0.3) | 34.4 (1.0) |
| | Outlier Exposure | 63.5 | 77.9 | 75.2 | 94.0 | 32.7 |
| | Energy Fine-Tuning | **57.8** | **74.6** | 77.3 | 94.7 | 34.3 |
| | DCM-Softmax (ours) | **58.0 (1.7)** | 79.3 (2.4) | **80.8 (1.2)** | **95.3 (0.3)** | 44.3 (2.1) |
| | DCM-Energy (ours) | **60.3 (2.8)** | 80.4 (1.6) | **81.0 (1.5)** | **95.3 (0.4)** | **47.6 (2.5)** |
| ID = CIFAR-10 OOD = CIFAR-100 | MSP | 45.7 (2.5) | 81.0 (5.6) | 86.8 (0.3) | 96.8 (0.1) | 53.4 (1.0) |
| | ODIN | 54.8 (4.6) | 85.1 (3.8) | 87.0 (0.4) | 96.6 (0.2) | 59.9 (0.9) |
| | Mahalanobis | 65.4 (2.2) | 85.0 (1.1) | 79.4 (1.0) | 94.7 (0.3) | 44.2 (2.1) |
| | Energy Score | 59.6 (2.6) | 89.0 (1.4) | 86.2 (0.4) | 96.2 (0.2) | 59.2 (0.5) |
| | Outlier Exposure | **28.3** | **57.9** | 93.1 | 98.5 | 76.5 |
| | Energy Fine-Tuning | 29.0 | 63.4 | **94.0** | **98.6** | **81.6** |
| | DCM-Softmax (ours) | 57.5 (6.1) | 90.0 (2.8) | 87.6 (0.7) | 96.5 (0.4) | 63.1 (0.7) |
| | DCM-Energy (ours) | 60.4 (5.3) | 90.5 (2.6) | 87.0 (0.9) | 96.3 (0.4) | 64.3 (1.2) |

Table 8: OOD detection performance with a WideResNet-40-2 model on CIFAR-10 to CIFAR-100 and CIFAR-100 to CIFAR-10. Bold numbers represent superior results. Numbers in parenthesis represent the standard deviation over 5 seeds. ↓: lower is better, ↑: higher is better.

# E   Semi-supervised novelty detection setting

For the sake of fair comparison, we also compare our algorithm's performance to binary classifier and ERD (Tifrea et al., 2022). These methods leverage an uncertainty dataset that contains both ID and OOD examples drawn from the distribution that we will see during test-time.

- **ERD**: Generates an ensemble by fine-tuning an ID pre-trained network on a combined ID + uncertainty dataset (which is a mixture of ID and OOD examples and given one label for all examples). Uses an ID validation set to early stop, and then uses the disagreement score between the networks on the ensemble to separate ID and OOD examples.

- **Binary Classifier**: The approach learns to discriminate between labeled ID set and uncertainty ID-OOD mixture set, with regularizations to prevent the entire uncertainty dataset to be classified as OOD.

We use the same datasets as Appendix D.

## E.1   Architecture and training details

- **Architecture**: For all experiments in this section, we use a ResNet-18 (He et al., 2015) network, same as Tifrea et al. (2022).

- **Hyper-parameters**: For ERD and binary classifier, we use the hyper-parameters and learning rate schedule used by Tifrea et al. (2022). For ERD, we standardize the experiments by using ensemble size = 3 for all experiments. The ensemble models are initialized with weights pre-trained solely on the ID training set for 100 epochs, and then each is further trained for 10 epochs. For binary classifier, we train all the networks from scratch for 100 epochs with a learning rate schedule described by Tifrea et al. (2022). For our method, we use the same hyper-parameters as Appendix D.

We use the same dataset splits, augmentations, uncertainty and test datasets as Appendix D.

| ID Dataset / Network | Method | SVHN | | LSUN (Crop) | | iSUN | | Texture | | Places365 | |
|---|---|---|---|---|---|---|---|---|---|---|---|
| | | AUROC (↑) | FPR@95 (↓) | AUROC (↑) | FPR@95 (↓) | AUROC (↑) | FPR@95 (↓) | AUROC (↑) | FPR@95 (↓) | AUROC (↑) | FPR@95 (↓) |
| CIFAR-10 ResNet-18 | VOS | 96.37 | 15.69 | 93.82 | 27.64 | 94.87 | 30.42 | 93.68 | 32.68 | 91.78 | 37.95 |
| | NPOS | 97.64 | 5.61 | 97.52 | 4.08 | 94.92 | 14.13 | 94.67 | 8.39 | 91.35 | 18.57 |
| | DCM-Softmax | 99.7 (0.1) | 0.4 (0.3) | 98.6 (0.8) | 6.6 (3.0) | 99.7 (0.1) | 0.6 (0.2) | 97.1 (0.2) | 14.8 (0.3) | 92.4 (0.3) | 32.6 (2.1) |
| | DCM-MaxLogit | 99.8 (0.1) | 0.3 (0.1) | 98.7 (0.7) | 6.0 (3.0) | 99.8 (0.1) | 0.5 (0.2) | 97.1 (0.2) | 14.9 (0.4) | 92.5 (0.3) | 34.4 (2.1) |
| | DCM-Energy | 99.8 (0.1) | 0.1 (0.1) | 98.8 (0.7) | 5.3 (3.6) | 99.8 (0.1) | 0.1 (0.1) | 97.1 (0.2) | 16.1 (1.1) | 92.5 (0.3) | 35.6 (2.2) |
| CIFAR-100 ResNet-34 | VOS | 73.11 | 78.50 | 85.72 | 59.05 | 82.66 | 72.45 | 80.08 | 75.35 | 75.85 | 84.55 |
| | NPOS | 97.84 | 11.14 | 82.43 | 56.27 | 85.48 | 51.72 | 92.44 | 35.20 | 71.30 | 79.08 |
| | Dream-OOD | 87.01 | 58.75 | 95.23 | 24.25 | 99.73 | 1.10 | 88.82 | 46.60 | 79.94 | 70.85 |
| | DCM-Softmax | 99.3 (0.2) | 1.8 (0.9) | 98.6 (0.3) | 8.7 (1.9) | 99.3 (0.2) | 2.3 (1.4) | 88.5 (0.5) | 46.6 (2.7) | 78.6 (0.4) | 67.7 (2.3) |
| | DCM-MaxLogit | 99.3 (0.2) | 2.0 (1.1) | 98.7 (0.2) | 7.3 (1.9) | 99.3 (0.2) | 2.3 (1.4) | 88.8 (0.5) | 46.9 (3.0) | 78.6 (0.4) | 68.8 (1.8) |
| | DCM-Energy | 99.2 (0.2) | 2.2 (1.2) | 98.9 (0.2) | 5.2 (2.0) | 99.4 (0.2) | 1.9 (0.8) | 89.3 (0.6) | 48.6 (3.3) | 78.3 (0.6) | 68.9 (1.9) |

Table 9: Comparison to methods that use synthetic outliers to make the model better at OOD detection. The table reports results for CIFAR-10 and CIFAR-100 as ID datasets. The results for CIFAR-10 are copied directly from Tao et al. (2023), and those for CIFAR-100 are copied directly from Du et al. (2023).

# F   Near-OOD Detection Setting

## F.1   Architecture and training details

- **Datasets**: Similar to Tifrea et al. (2022), we try two settings: (1) ID = first 5 classes of CIFAR-10, OOD = last 5 classes of CIFAR-100, (2) ID = first 50 classes of CIFAR-100, OOD = last 50 classes of CIFAR-100.

- **Dataset splits**: We use 20,000 train and 5,000 validation label-balanced images during training.

- **Uncertainty and test split construction**: We use two disjoint datasets of size 3,000 as uncertainty and test datasets. Each dataset contains 2,500 ID and 500 OOD examples.

We use the same architecture, hyper-parameters and augmentations, as Appendix E.

# G   Transductive OOD Detection Setting

In scenarios where examples similar to those encountered at test time are not available, we can use a modified version of DCM in which we use an uncertainty dataset consisting of the test set itself. We expect this transductive variant to perform slightly worse since we end up directly minimizing confidence on ID test examples, in addition to the general absence of information from additional unlabeled data. We assess the performance of DCM in this transductive setting. In Table 5 and Table 6, we compare the performance of DCM in this transductive setting to the regular setting. While we observe a slight drop compared to the default DCM, we still show competitive performance in the transductive setting compared to prior approaches, as shown in Table 7.

# H   Comparison to Methods that Use Synthetic Outliers

Since DCM uses a mixture of known and unknown samples to finetune the model to be more conservative during test-time, it is reasonable to compare DCM with OOD detection methods that generate synthetic outliers and use them to make the model conservative. In this section, we compare against SOTA methods of this type: VOS (Du et al., 2022a), NPOS (Tao et al., 2023) and Dream-OOD (Du et al., 2023). We report results for CIFAR-10 and CIFAR-100 as ID datasets. We use SVHN (Netzer et al., 2011), LSUN (Yu et al., 2015; Liang et al., 2017a), iSUN (Xu et al., 2015; Liang et al., 2017a), Texture (Cimpoi et al., 2013) and Places365 (Zhou et al., 2017) as OOD datasets, following earlier work. Results in Table 9 show that DCM outperforms these points of comparison in most cases. This suggests that directly using outlier inputs similar to those seen during test time, instead of synthesizing such outlier examples, is an effective approach.

| Methods | iNaturalist | | SUN | | Places | | Texture | |
|---|---|---|---|---|---|---|---|---|
| | FPR@95 ($\downarrow$) | AUROC ($\uparrow$) | FPR@95 ($\downarrow$) | AUROC ($\uparrow$) | FPR@95 ($\downarrow$) | AUROC ($\uparrow$) | FPR@95 ($\downarrow$) | AUROC ($\uparrow$) |
| MCM (zero-shot) | 32.08 | 94.41 | 39.21 | 92.28 | 44.88 | 89.83 | 58.05 | 85.96 |
| (Fine-tuned) | | | | | | | | |
| Fort et al./MSP | 54.05 | 87.43 | 73.37 | 78.03 | 72.98 | 78.03 | 68.85 | 79.06 |
| ODIN | 30.22 | 94.65 | 54.04 | 87.17 | 55.06 | 85.54 | 51.67 | 87.85 |
| Energy | 29.75 | 94.68 | 53.18 | 87.33 | 56.40 | 85.60 | 51.35 | 88.00 |
| GradNorm | 81.50 | 72.56 | 82.00 | 72.86 | 80.41 | 73.70 | 79.36 | 70.26 |
| ViM | 32.19 | 93.16 | 54.01 | 87.19 | 60.67 | 83.75 | 53.94 | 87.18 |
| KNN | 29.17 | 94.52 | 35.62 | 92.67 | 39.61 | 91.02 | 64.35 | 85.67 |
| VOS | 31.65 | 94.53 | 43.03 | 91.92 | 41.62 | 90.23 | 56.67 | 86.74 |
| VOS+ | 28.99 | 94.62 | 36.88 | 92.57 | 38.39 | 91.23 | 61.02 | 86.33 |
| NPOS | 16.58 | 96.19 | 43.77 | 90.44 | 45.27 | 89.44 | 46.12 | 88.80 |
| DCM-Softmax | 2.6 (0.5) | 99.2 (0.1) | 32.9 (1.5) | 94.2 (0.2) | 35.9 (1.8) | 93.8 (0.3) | 11.2 (1.0) | 97.9 (0.1) |
| DCM-MaxLogit | 1.8 (0.4) | 99.4 (0.1) | 27.5 (1.4) | 94.9 (0.2) | **32.5 (2.8)** | 94.5 (0.3) | 8.2 (0.8) | 98.3 (0.1) |
| DCM-Energy | **0.5 (0.2)** | **99.6 (0.1)** | **24.5 (1.7)** | **95.8 (0.2)** | 30.8 (3.0) | **95.4 (0.3)** | **4.3 (0.6)** | **98.8 (0.1)** |

Table 10: OOD detection performance for ImageNet-1K as ID dataset, using a ViT-B/16 model. The performance metrics for baselines are copied directly from Tao et al. (2023). We report the mean and standard error over 3 seeds for DCM.

# I OOD Detection Evaluation on ImageNet-1K (Deng et al., 2009)

In order to show that DCM scales to large scale datasets beyond CIFAR-10 and CIFAR-100, we also report results on ImageNet-1K.

## I.1 OOD datasets

We use iNaturalist (Horn et al., 2018), SUN (Xiao et al., 2010), Texture (Cimpoi et al., 2013) and Places (Zhou et al., 2017) as OOD datasets following earlier work such as (Tao et al., 2023). Particularly, we use the subsets of these datasets chosen by (Tao et al., 2023) for fair comparison.

## I.2 Baselines

We compare DCM with the following baselines:

- Maximum concept matching (MCM) (Ming et al., 2022)

- Maximum softmax probability (Hendrycks & Gimpel, 2016; Fort et al., 2021)

- ODIN score (Liang et al., 2017a)

- Energy score (Liu et al., 2020)

- Grad norm (Huang et al., 2021)

- ViM (Wang et al., 2022)

- KNN distance (Sun et al., 2022)

- Virtual Outlier Synthesis (VOS) (Du et al., 2022a)

- Non-parametric Outlier Synthesis (NPOS) (Tao et al., 2023)

## I.3 Training details

For these experiments, we use a ViT-B/16 model (Dosovitskiy et al., 2021) pretrained on ImageNet-1K. For each OOD dataset, we run DCM using an uncertainty dataset of size 25000, with 20000 class balanced examples from ImageNet validation set (ID) and 5000 examples from the OOD dataset. We test on a dataset

| OOD Dataset | MSP (Weight Decay) | | | | | | DCM-Softmax |
|---|---|---|---|---|---|---|---|
| | 0.0005 | 0.0006 | 0.0008 | 0.001 | 0.005 | 0.01 | |
| SVHN | 77.7 | 81.9 | 78.3 | 76.5 | 60.2 | 53.4 | 99.7 |
| LSUN | 68.5 | 69.5 | 71.0 | 73.5 | 66.4 | 68.2 | 99.5 |
| TinyImageNet | 68.0 | 66.5 | 69.8 | 72.4 | 61.2 | 61.4 | 98.7 |
| iSUN | 67.1 | 68.5 | 68.4 | 71.5 | 61.5 | 63.5 | 99.1 |

Table 11: OOD detection AUROC of MSP (ID: CIFAR-100) when the classifier (WideResNet-40-2) has been pre-trained with different weight decays. We see that controlling for weight decay can improve OOD detection performance slightly. Best performance of MSP for each OOD data is marked with an underline.

of size 25000, with the same composition as the uncertainty dataset but containing different images than that of the uncertainty dataset. Additionally, we use the remaining 10000 class balanced images from the validation set, not used in the uncertainty dataset or the test dataset, as our validation set. For DCM, we fine-tune the pre-trained model for an additional 5 epochs, with AdamW optimizer, learning rate $3 \times 10^{-5}$, weight decay 0.01 and use cosine annealing learning rate decay. Similar to our experiments on CIFAR-10 and CIFAR-100, we use confidence weight $\lambda = 0.5$. In each batch, the model sees 32 ID training image and 64 uncertainty dataset image, and the epoch ends when we exhaust the entire uncertainty dataset.

### I.4   Results

Table 10 shows the performance of DCM compared to the baselines. We see that DCM outperform all other methods, and achieving state-of-the-art results on all 4 OOD datasets, showing that DCM also scale to large-scale datasets.

## J   DCM for OOD Detection Preserves ID Performance

DCM does slightly degrade ID performance, as seen in Tables 2 to 4, but this degradation is small compared to the gains achieved in OOD detection. One can also use DCM to separate OOD examples, and then use the pre-trained ERM weights for the ID classification task to get the best of both worlds. DCM preserves ID performance on ImageNet-1K as well, showing that this property scales to much larger datasets.

## K   Exploring DCM in the OOD Detection Setting

**Effect of number of epochs in the second fine-tuning stage** Since the second fine-tuning stage is the crucial step for our algorithm, we try different number of epochs for this stage and see its effect. Figure 5 shows the results. We see that the performance variation due to varying the number of epochs is negligible, implying DCM is robust to the choice of this hyper-parameters. We also see that in the 4/5 cases we have tried out, our default choice of 10 for the number of fine-tuning epochs do not achieve the best performance, justifying our experiment design.

**DCM compared to other forms of regularization.** For the sake of completeness, we compare DCM with weight decay and label smoothing (Szegedy et al., 2015), two popular regularization method. In these experiments, we train a WideResNet-40-2 model on CIFAR-100 by varying each regularization factor during training the model from scratch while keeping every other hyper-parameter and training details fixed at those described in Appendix D.4. Tables 11 and 12 show the results of these experiments.

We observe that controlling weight decay can result in a better OOD detection performance compared to the default weight decay of 0.0005 we used in this paper. However, this is not true for label smoothing, since using label smoothing = 0.0 achieves the best OOD detection performance in all experiments. Overall, DCM performs much better, showing it is an effective form of regularization against out-of-distribution samples.

| OOD Dataset | MSP (Label smoothing) | | | | | DCM-Softmax |
|---|---|---|---|---|---|---|
| | 0 | 0.25 | 0.5 | 0.75 | 1 | |
| SVHN | 77.7 | 74.9 | 67.2 | 76.4 | 50.0 | 99.7 |
| LSUN | 68.5 | 56.8 | 61.7 | 63.3 | 50.0 | 99.5 |
| TinyImageNet | 68.0 | 59.4 | 62.4 | 58.5 | 50.0 | 98.7 |
| iSUN | 67.1 | 55.1 | 61.0 | 59.2 | 50.0 | 99.1 |

Table 12: OOD detection AUROC of MSP (ID: CIFAR-100) when the classifier (WideResNet-40-2) has been pre-trained with label smoothing. Increasing label smoothing hurts OOD detection performance, as the model trained with label smoothing 0 does the best on all 4 OOD datasets. Best performance of MSP for each OOD data is marked with an underline.

| ID Dataset | OOD Dataset | Classification Accuracy | | OOD Detection AUROC | | OOD Detection FPR@95 | |
|---|---|---|---|---|---|---|---|
| | | Fine-tune | Pre-train | Fine-tune | Pre-train | Fine-tune | Pre-train |
| CIFAR-10 | SVHN | 90.6 | 93.7 | 99.7 | 99.7 | 0.8 | 0.4 |
| | TinyImageNet | 90.4 | 93.5 | 99.3 | 99.3 | 2.2 | 2.6 |
| | LSUN | 90.6 | 93.7 | 99.2 | 99.8 | 2.5 | 0.5 |
| | iSUN | 90.2 | 93.5 | 99.5 | 99.7 | 1.4 | 0.6 |
| CIFAR-100 | SVHN | 69.3 | 71.4 | 99.6 | 99.6 | 0.4 | 0.6 |
| | TinyImageNet | 68.1 | 71.1 | 99.0 | 98.7 | 3.2 | 5.9 |
| | LSUN | 69.0 | 71.0 | 99.7 | 99.5 | 0.7 | 1.1 |
| | iSUN | 68.1 | 71.2 | 99.4 | 99.1 | 2.0 | 2.7 |

Table 13: Comparison between DCM used during pre-training a model from scratch and fine-tuning a pre-trained model. In both cases, we use the maximum softmax probability for OOD detection, and use a WideResNet-40 model.

**DCM used during pre-training.** We have so far used DCM as a fine-tuning mechanism on top of a pre-trained model. Here we explore DCM as pre-training algorithm. Specifically, we skip pre-training on the ID dataset from Algorithm 2, and directly train a model from scratch using objective (3), i.e., the weighted sum of cross-entropy loss on the ID training set and the confidence loss on the unlabeled set. We use WideResNet-40-2 models and train for 100 epochs. Since the uncertainty dataset is much smaller than the ID training set, we repeat the images from the uncertainty dataset so that the model sees the entire training set during each epoch. The dataset composition and other hyper-parameters are same as Appendix D.

Table 13 shows the results of these experiments. Training directly from scratch using DCM leads to slightly lower ID accuracy but comparable performance on OOD detection tasks. The loss in ID performance is due to the optimization task of DCM being harder: it has two components, namely the cross-entropy loss on the ID training set and confidence minimization loss on ID examples of the uncertainty dataset, that work in opposite directions, making learning the ID classification task harder. Whereas if we fine-tune an ID pre-trained model, the model already has learned the ID task and the fine-tuning step only further modifies the decision boundary, maintaining higher ID performance.

We use DCM to fine-tune a pretrained model due to computational reasons: the fine-tuning step is much shorter than the pre-training step, and the ratio of compute required becomes smaller when we use larger and larger datasets. When we pre-train one epoch using DCM, the model essentially sees 2x the number of images compared to regular pre-training and hence takes 2x compute, since it sees equal number of images from the training and auxiliary datasets. Also this lets us take one ID pre-trained model and adapt it relatively quickly to different test distributions, avoiding the costly pre-training process for each different test distribution.

# L  Selective Classification Experiment Details

## L.1  Baselines

- **MSP** (Hendrycks & Gimpel, 2016): Also referred to as Softmax Response (SR), MSP directly uses the maximum softmax probability assigned by the model as an estimate of confidence. MSP has been shown to distinguish in-distribution test examples that the model gets correct from the ones that it gets incorrect.

- **MaxLogit** (Hendrycks et al., 2022): Directly uses the maximum logit outputted by the model as an estimate of confidence.

- **Ensemble** (Lakshminarayanan et al., 2017): Uses an ensemble of 5 models, each trained with ERM on the ID train distribution with different random seeds. Following Lakshminarayanan et al. (2017), we use the entropy of the average softmax predictions of the models in the ensemble as the disagreement metric.

- **Binary Classifier** (Kamath et al., 2020): Trains a classifier on the labeled training and validation sets to predict inputs for which the base model will be correct versus incorrect. The classifier takes as input the softmax probabilities outputted by the base model. For the Binary Classifier, we found the MLP with softmax probabilities to work best compared to a random forest classifier and MLP with last-layer features.

- **Fine-tuning**: First trains a model on the training set, then fine-tunes the model on the validation set.

- **Deep Gamblers** (Liu et al., 2019): Trains a classifier using a loss function derived from the doubling rate in a hypothetical horse race. Deep Gamblers introduces an extra $(c + 1)$-th class that represents abstention. Minimizing this loss corresponds to maximizing the return, where the model makes a bet or prediction when confident, and abstains when uncertain.

- **Self-Adaptive Training** (Huang et al., 2020): Trains a classifier using model predictions to dynamically calibrate the training process. SAT introduces an extra $(c + 1)$-th class that represents abstention and uses training targets that are exponential moving averages of model predictions and ground-truth targets.

## L.2  Datasets

- **CIFAR-10 (Krizhevsky et al., a) → CIFAR-10-C** (Hendrycks & Dietterich, 2019): The task is to classify images into 10 classes, where the target distribution contains severely corrupted images. We run experiments over 15 corruptions (brightness, contrast, defocus blur, elastic transform, fog, frost, gaussian noise, glass blur, impulse noise, jpeg compression, motion blur, pixelate, shot noise, snow, zoom blur) and use the data loading code from Croce et al. (2020).

- **Waterbirds** (Wah et al., 2011; Sagawa et al., 2019): The Waterbirds dataset consists of images of landbirds and waterbirds on land or water backgrounds from the Places dataset (Zhou et al., 2017). The train set consists of 4,795 images, of which 3,498 are of waterbirds on water backgrounds, and 1,057 are of landbirds on land backgrounds. There are 184 images of waterbirds on land and 56 images of landbirds on water, which are the minority groups.

- **Camelyon17** (Koh et al., 2021; Bandi et al., 2018): The Camelyon17 dataset is a medical image classification task from the WILDS benchmark (Koh et al., 2021). The dataset consists of $450,000$ whole-slide images of breast cancer metastases in lymph node from 5 hospitals. The input is a $96 \times 96$ image, and the label $y$ indicates whether there is a tumor in the image. The train set consists of lymph-node scans from 3 of the 5 hospitals, while the OOD validation set and OOD test datasets consists of lymph-node scans from the fourth and fifth hospitals, respectively.

- **FMoW** (Koh et al., 2021): The FMoW dataset is a satellite image classification task from the WILDS benchmark (Koh et al., 2021). The dataset consists of satellite images in various geographic locations from $2002 - 2018$. The input is a $224 \times 224$ RGB satellite image, and the label $y$ is one of 62 building or land use categories. The train, validation, and test splits are based on the year that the images were taken: the train, ID validation, and ID test sets consist of images from $2002 - 2013$, the OOD validation set consists of images from $2013 - 2016$, and the OOD test set consists of images from $2016 - 2018$.

### L.3  CIFAR-10 → CIFAR-10-C training details

- **Architecture:** We use a WideResNet-28-10 trained on the source CIFAR-10 distribution and attains 94.78% clean accuracy (Croce et al., 2020). The base models for Deep Gamblers and Self-Adaptive Training use the same architecture with an additional 11th class in the final linear layer.

- **Hyper-parameters**: DCM, Fine-tuning, Binary Classifier, Deep Gamblers, and Self-Adaptive Training fine-tune the base models for 10 epochs. We perform hyperparameter tuning on a separate, held-out ID validation set. We tune all baselines over the learning rates {1e-3, 1e-4, 1e-5}, and use an initial learning rate of 1e-3 for all baselines except DCM, which uses an initial learning rate of 1e-4. For DCM, we use a confidence weight of $\lambda = 0.5$ for all corruptions, as in Hendrycks et al. (2018). For Deep Gamblers, we use a reward of 3.2 and tune over rewards in the range $[2.0, 4.2]$ with a step size of 0.2, as in Liu et al. (2019). We use SGD with a cosine learning rate schedule, weight decay of $5 \times 10^{-4}$, and batch sizes of 128 and 256 for the fine-tuning set and uncertainty datasets, respectively.

- **Validation and test set construction:** We use the CIFAR-10 test set, and split it into a validation set of 5000 images, a test set of 4000 images, and set aside 1000 images for hyperparameter tuning. Similarly, for CIFAR-10-C, we use a validation set of 5000 images, and a test set of 4000 images.

  Each of our settings merges the train/val/test splits from the corresponding datasets. For example, Val = CIFAR-10, Test = CIFAR-10 + CIFAR-10-C uses a validation set of 5000 CIFAR-10 images for fine-tuning and a test set of 4000 CIFAR-10 and 4000 CIFAR-10-C images. Note that our combined CIFAR-10 + CIFAR-10-C test sets have a 1:1 clean-to-corrupted ratio.

- **Augmentations:** For DCM and fine-tuning, we use the same standard random horizontal flip and random crop $(32 \times 32)$.

### L.4  Waterbirds training details

- **Architecture:** For our base model, we train a pretrained ResNet50 from `torchvision` on a subset of the Waterbirds train set (details of the split are described below). We follow the training details for the ERM baseline used by Sagawa et al. (2019), and use SGD with a momentum term of 0.9, batch normalization, and no dropout. We use a fixed learning rate of 0.001, a $\ell_2$ penalty of $\lambda = 0.0001$ and train for 300 epochs.

- **Hyper-parameters**: DCM, Fine-tuning, Binary Classifier, Deep Gamblers, and Self-Adaptive Training fine-tune the base models for 20 epochs. We perform hyperparameter tuning on a separate, held-out ID validation set. We tune all baselines over the learning rates {1e-3, 1e-4, 1e-5}, and use an initial learning rate of 1e-3 for all baselines except DCM, which uses an initial learning rate of 1e-4. For DCM, we use a confidence weight of $\lambda = 0.5$. For Deep Gamblers, we use a reward of 1.4 and tune over rewards in the range $[1.0, 2.0]$ with a step size of 0.2. We use SGD with a cosine learning rate schedule, weight decay of $5 \times 10^{-4}$, and batch sizes of 64 and 128 for the fine-tuning set and uncertainty datasets, respectively.

- **Validation and test set construction:** We split the Waterbirds train set from Sagawa et al. (2019) into two sets, one which we use to pretrain a base ERM model, and the other which we use as our ID validation set. We maintain group ratios, and the ID train and validation sets each contain 2,397 images, consisting of 1749 waterbirds on water, 528 landbirds on land, 92 waterbirds on land, and 28 landbirds on water. Our test set is the same test set from Sagawa et al. (2019).

- **Augmentations:** DCM and Fine-Tuning use the train augmentations as in Sagawa et al. (2019): a random resized crop $(224 \times 224)$ and random horizontal flip.

### L.5  Camelyon17 training details

- **Architecture:** We use a DenseNet121 pre-trained on the Camelyon17 train set from Koh et al. (2021) as our base model. These models are trained for 5 epochs with a learning rate of 0.001, $\ell_2$ regularization strength of 0.01, batch size of 32, and SGD with momentum set to 0.9.

- **Hyper-parameters**: DCM, Fine-tuning, Binary Classifier, Deep Gamblers, and Self-Adaptive Training fine-tune the base models for 1 epoch. We perform hyperparameter tuning on a separate, held-out ID validation set. We tune all baselines over the learning rates {1e-3, 1e-4, 1e-5}, and use an initial learning rate of 1e-4 for all baselines. For DCM, we use a confidence weight of $\lambda = 0.5$. For Deep Gamblers, we use a reward of 1.4 and tune over rewards in the range $[1.0, 2.0]$ with a step size of 0.2. We use an AdamW optimizer with a cosine learning rate schedule, weight decay of $5 \times 10^{-4}$, and batch size of 32 for the fine-tuning set and uncertainty datasets.

- **Validation and test set construction:** We use the train / ID val / OOD val / OOD test splits from the WILDS benchmark to construct our validation and test sets. For our ID validation set and ID test set, we split the Camelyon17 ID validation set into two equally-sized subsets and maintain group ratios. The Camelyon17 ID validation consists of samples from the same 3 hospitals as the train set. We use the OOD test set as our target distribution, which contains samples from the 5th hospital.

- **Augmentations:** DCM and Fine-tuning use random horizontal flips.

### L.6 FMoW training details

- **Architecture:** We use FMoW ERM models from the WILDS benchmark (Koh et al., 2021) as our base model. These models use DenseNet121 pretrained on ImageNet with no $\ell_2$ regularization, Adam optimizer with an initial learning rate of 1e-4 that decays by 0.96 per epoch, and train for 50 epochs with early stopping and batch size of 64.

- **Hyper-parameters**: DCM, Fine-tuning, Binary Classifier, Deep Gamblers, and Self-Adaptive Training fine-tune the base models for 5 epochs. We perform hyperparameter tuning on a separate, held-out ID validation set. We tune all baselines over the learning rates {1e-3, 1e-4, 1e-5, 1e-6}, and use an initial learning rate of 1e-3 for SAT, 1e-4 for Fine-tuning, Deep Gamblers, and Binary Classifier, and 1e-5 for DCM. For DCM, we use a confidence weight of $\lambda = 0.1$ and tune over the weights $\{0.01, 0.05, 0.1, 0.5, 1.0, 1.5\}$ on a held-out ID validation set. For Deep Gamblers, we use a reward of 35 and tune over rewards in the range $[5.0, 65.0]$ with a step size of 5.0. We use an AdamW optimizer with a cosine learning rate schedule, weight decay of $5 \times 10^{-4}$, and batch sizes of 16 and 32 for the uncertainty dataset and fine-tuning sets, respectively.

- **Validation and test set construction:** We use the train, ID validation, OOD validation, and OOD test splits from the WILDS benchmark as our validation and test sets. Specifically, we use the ID validation set, ID test set, and OOD test sets. For example, the task Val = FMoW ID, Test = FMoW ID + FMoW OOD uses the WILDS ID val set for validation, and the WILDS ID and OOD test sets for testing.

- **Augmentations:** DCM and Fine-tuning use random horizontal flips.

## M  Selective Classification Ablations

### M.1  Effect of Validation Set Size

Figure 7 shows the performance of DCM on selective classification when we vary the size of the validation set. DCM consistently outperforms baselines in distribution-shift settings with different validation set sizes.

### M.2  DCM compared to other regularization methods

Table 14 shows the comparison between pre-training with label smoothing (Szegedy et al., 2015) as a form of regularization and DCM. We see that higher weights of label smoothing degrades performance, and therefore is not an alternative to DCM.

## N  Compute

All model training and experiments were conducted on a single NVIDIA RTX Titan or A40 GPU.

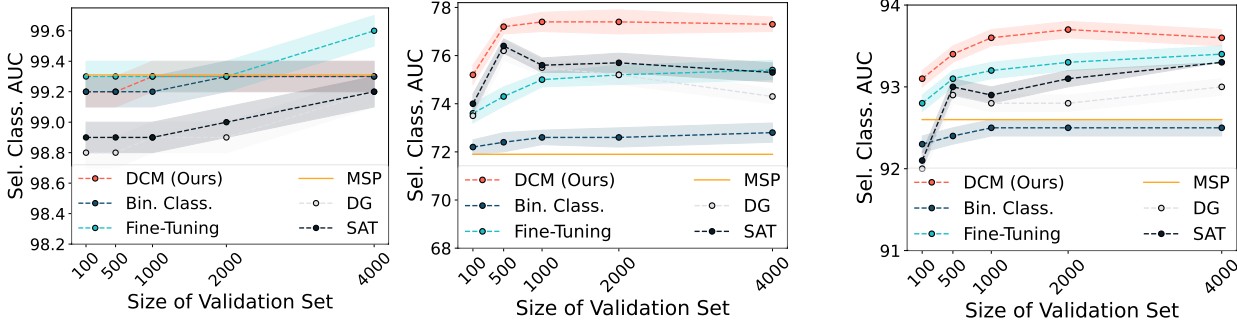

Figure 7: Selective classification performance of DCM when we vary the size of the validation set. Left: CIFAR-10 → CIFAR-10, Middle: CIFAR-10 → CIFAR-10-C, Right: CIFAR-10 → CIFAR-10 + CIFAR-10-C. DCM consistently outperforms baselines in distribution-shift settings with different validation set sizes.

| | MSP with Label Smoothing | | | | | DCM-Softmax |
|---|---|---|---|---|---|---|
| Eval Data | 0 | 0.25 | 0.5 | 0.75 | 1 | |
| ID | 81.3 | 81.0 | 80.7 | 71.0 | 59.6 | **82.9** |
| ID+OOD | 77.1 | 76.9 | 76.4 | 67.8 | 52.5 | **78.9** |
| OOD | 74.5 | 74.3 | 74.2 | 64.0 | 55.1 | **76.4** |

Table 14: Pre-training with label smoothing does not improve selective classification. Here, we compare an MSP classifier pre-trained with varying degrees of label smoothing with DCM on the FMoW dataset. Higher weights of label smoothing degrade performance. We underline the best performance of MSP with label smoothing.

| Method | ECE (↓) | Acc (↑) | AUC (↑) | Acc@90 (↑) | Acc@95 (↑) | Acc@99 (↑) | Cov@90 (↑) | Cov@95 (↑) | Cov@99 (↑) |
|---|---|---|---|---|---|---|---|---|---|
| Val = CIFAR-10, Test = CIFAR-10 | | | | | | | | | |
| MSP | 0.5 (0.1) | 95.2 (0.1) | 99.3 (0.1) | 98.4 (0.1) | 97.2 (0.1) | 95.7 (0.1) | **100 (0.0)** | **100 (0.0)** | 87.0 (0.1) |
| MaxLogit | 0.6 (0.1) | 95.1 (0.1) | 98.9 (0.1) | 97.9 (0.1) | 96.8 (0.1) | 95.6 (0.1) | **100 (0.0)** | **100 (0.0)** | 80.4 (0.1) |
| Ensemble | 0.6 (0.1) | 96.1 (0.1) | 99.5 (0.1) | 98.9 (0.1) | 97.6 (0.1) | 96.1 (0.1) | **100 (0.0)** | **100 (0.0)** | 86.5 (0.1) |
| Binary Classifier | 1.4 (0.1) | 95.2 (0.1) | 99.3 (0.1) | 98.4 (0.1) | 97.2 (0.2) | 95.7 (0.2) | **100 (0.0)** | **100 (0.0)** | 87.0 (2.3) |
| Fine-Tuning | **0.3 (0.2)** | **96.2 (0.1)** | **99.6 (0.1)** | **99.1 (0.2)** | **98.7 (0.1)** | **97.5 (0.2)** | **100 (0.0)** | **100 (0.0)** | **91.6 (0.9)** |
| DG | 0.8 (0.1) | 94.5 (0.0) | 99.0 (0.0) | 97.4 (0.1) | 96.4 (0.0) | 94.9 (0.1) | **100 (0.0)** | 98.7 (0.1) | 76.2 (1.7) |
| SAT | 0.7 (0.1) | 94.7 (0.0) | 99.2 (0.0) | 97.6 (0.1) | 96.3 (0.0) | 94.9 (0.1) | **100 (0.0)** | 98.8 (0.1) | 81.6 (1.1) |
| DCM (ours) | 1.0 (0.2) | 94.7 (0.2) | 99.2 (0.2) | 98.0 (0.2) | 96.5 (0.2) | 94.8 (0.2) | **100 (0.0)** | 98.6 (0.4) | 83.9 (1.0) |
| Val = CIFAR-10, Test = CIFAR-10 + CIFAR-10-C | | | | | | | | | |
| MSP | 9.3 (0.1) | 75.8 (0.1) | 92.6 (0.1) | 80.4 (0.1) | 78.3 (0.1) | 76.4 (0.1) | 72.4 (0.2) | 60.6 (0.2) | 27.4 (0.7) |
| MaxLogit | 9.4 (0.0) | 75.7 (0.1) | 91.7 (0.0) | 80.4 (0.0) | 78.2 (0.0) | 76.3 (0.0) | 70.5 (0.1) | 54.0 (0.3) | 10.1 (0.3) |
| Ensemble | 8.4 (0.1) | 76.4 (0.1) | 93.4 (0.1) | 81.2 (0.1) | 78.8 (0.1) | 76.8 (0.1) | 72.9 (0.2) | 60.5 (0.3) | 35.0 (0.6) |
| Binary Classifier | **7.9 (0.1)** | 75.4 (0.1) | 92.5 (0.1) | 80.3 (0.1) | 78.1 (0.1) | 76.2 (0.1) | 72.0 (0.2) | 59.9 (0.3) | 30.5 (1.7) |
| Fine-Tuning | 8.2 (0.1) | 75.2 (0.1) | 93.4 (0.1) | 81.3 (0.1) | 78.9 (0.1) | 77.0 (0.1) | 74.2 (0.2) | 63.4 (0.3) | **42.4 (0.9)** |
| DG | 8.2 (0.0) | 76.0 (0.1) | 93.0 (0.1) | 81.0 (0.0) | 78.8 (0.0) | 76.9 (0.0) | 73.2 (0.2) | 60.7 (0.5) | 33.4 (1.2) |
| SAT | 7.8 (0.0) | 76.2 (0.1) | 93.3 (0.0) | 81.1 (0.0) | 78.7 (0.1) | 76.8 (0.0) | 74.1 (0.0) | 62.7 (0.1) | 42.7 (0.3) |
| DCM (ours) | 8.0 (0.1) | **77.0 (0.1)** | **93.6 (0.1)** | **82.0 (0.1)** | **79.7 (0.1)** | **77.7 (0.1)** | **75.2 (0.1)** | **63.8 (0.3)** | 43.6 (0.9) |
| Val = CIFAR-10, Test = CIFAR-10-C | | | | | | | | | |
| MSP | 13.8 (0.1) | 56.4 (0.1) | 70.1 (0.1) | 57.4 (0.2) | 56.0 (0.2) | 54.8 (0.1) | 30.2 (0.3) | 20.9 (0.5) | 7.6 (0.3) |
| MaxLogit | 14.6 (0.0) | 56.4 (0.1) | 71.7 (0.1) | 59.4 (0.1) | 58.0 (0.0) | 56.8 (0.0) | 23.2 (0.7) | 11.3 (0.3) | 3.5 (0.5) |
| Ensemble | 13.1 (0.1) | 57.2 (0.1) | 75.3 (0.1) | 61.8 (0.1) | 58.6 (0.1) | 57.9 (0.1) | 28.0 (0.2) | 17.6 (0.5) | 6.8 (0.2) |
| Binary Classifier | 13.6 (0.1) | 56.2 (0.2) | 72.8 (0.2) | 59.5 (0.2) | 58.0 (0.2) | 56.7 (0.6) | 28.5 (0.6) | 16.4 (0.8) | 8.2 (0.8) |
| Fine-Tuning | 12.7 (0.1) | 57.6 (0.1) | 75.4 (0.1) | 61.7 (0.2) | 60.2 (0.3) | 58.8 (0.3) | 33.6 (0.8) | 22.5 (0.8) | 8.6 (0.5) |
| DG | 12.7 (0.0) | 56.8 (0.1) | 74.3 (0.2) | 61.4 (0.1) | 59.9 (0.0) | 58.7 (0.0) | 28.4 (0.5) | 17.2 (0.5) | 7.2 (0.3) |
| SAT | 12.5 (0.0) | 57.4 (0.1) | 75.3 (0.1) | 61.4 (0.1) | 59.8 (0.1) | 58.4 (0.1) | 32.5 (0.5) | 22.0 (0.8) | 8.6 (0.2) |
| DCM (ours) | **12.3 (0.1)** | **59.4 (0.1)** | **77.5 (0.2)** | **64.1 (0.2)** | **62.4 (0.2)** | **61.0 (0.2)** | **37.6 (0.6)** | **25.2 (1.0)** | 8.9 (0.2) |

Table 15: Selective classification performance on various distribution shift tasks constructed from the CIFAR-10 and CIFAR-10-C datasets. Bold numbers represent superior results, and parentheses show the standard error over 3 random seeds. DCM consistently outperforms MSP, MaxLogit, Deep Gamblers (DG), and Self-Adaptive Training (SAT), and outperforms all 7 prior methods when the validation and test sets are from different distributions.

| Method | ECE (↓) | Acc (↑) | AUC (↑) | Acc@90 (↑) | Acc@95 (↑) | Acc@99 (↑) | Cov@90 (↑) | Cov@95 (↑) | Cov@99 (↑) |
|---|---|---|---|---|---|---|---|---|---|
| Val = Waterbirds-Train, Test = Waterbirds-Train | | | | | | | | | |
| MSP | 3.4 (0.0) | 96.8 (0.0) | **98.7 (0.0)** | 99.1 (0.0) | 98.2 (0.0) | 97.1 (0.0) | **100 (0.0)** | **100 (0.0)** | 90.1 (0.0) |
| MaxLogit | 3.2 (0.0) | 96.8 (0.0) | 98.6 (0.0) | 99.3 (0.0) | 98.2 (0.0) | 97.2 (0.0) | **100 (0.0)** | **100 (0.0)** | 91.0 (0.0) |
| Ensemble | 3.1 (0.0) | **97.0 (0.0)** | **98.7 (0.0)** | 98.9 (0.0) | 98.2 (0.0) | 97.2 (0.0) | **100 (0.0)** | **100 (0.0)** | 88.8 (0.0) |
| Binary Classifier | 3.4 (0.0) | 96.0 (0.0) | **98.7 (0.0)** | 99.1 (0.0) | 98.2 (0.0) | 97.1 (0.0) | **100 (0.0)** | **100 (0.0)** | 90.1 (0.0) |
| Fine-Tuning | 1.1 (0.0) | 96.9 (0.0) | **98.7 (0.0)** | **99.4 (0.0)** | **98.6 (0.0)** | 97.3 (0.0) | **100 (0.0)** | **100 (0.0)** | 91.8 (0.0) |
| DG | 1.3 (0.3) | **97.0 (0.0)** | 98.5 (0.0) | 98.8 (0.1) | 98.0 (0.1) | 97.3 (0.0) | **100 (0.0)** | **100 (0.0)** | 86.9 (0.5) |
| SAT | **0.7 (0.4)** | 96.8 (0.0) | 98.6 (0.0) | 99.1 (0.1) | 98.3 (0.1) | **97.4 (0.1)** | **100 (0.0)** | **100 (0.0)** | 91.3 (0.9) |
| DCM (ours) | 1.8 (0.6) | 96.8 (0.0) | **98.7 (0.0)** | 99.2 (0.0) | 98.3 (0.1) | 97.2 (0.0) | **100 (0.0)** | **100 (0.0)** | **91.9 (0.1)** |
| Val = Waterbirds-Train, Test = Waterbirds-Test | | | | | | | | | |
| MSP | 15.1 (0.0) | 84.3 (0.0) | 94.4 (0.0) | 88.2 (0.0) | 86.8 (0.0) | 85.3 (0.0) | 83.9 (0.0) | 60.9 (0.0) | 27.5 (0.0) |
| MaxLogit | 18.1 (0.0) | 84.3 (0.0) | 94.2 (0.0) | 87.9 (0.0) | 86.3 (0.0) | 84.7 (0.0) | 82.2 (0.0) | 60.9 (0.0) | 23.8 (0.0) |
| Ensemble | 14.9 (0.0) | 85.0 (0.0) | 94.4 (0.0) | 88.4 (0.0) | 87.0 (0.0) | 85.4 (0.0) | 85.0 (0.0) | 62.0 (0.0) | 25.6 (0.0) |
| Binary Classifier | 16.4 (0.3) | 84.9 (0.2) | 94.0 (0.2) | 87.5 (0.3) | 86.1 (0.2) | 84.8 (0.2) | 81.2 (1.8) | 59.8 (2.0) | 24.2 (1.1) |
| Fine-Tuning | 15.3 (0.4) | 85.9 (0.5) | 94.7 (0.2) | 89.0 (0.5) | 87.2 (0.5) | **86.2 (0.5)** | 86.8 (1.4) | 64.0 (2.7) | 27.9 (2.7) |
| DG | 17.3 (0.4) | 85.1 (0.1) | 94.8 (0.1) | 88.6 (0.2) | 87.0 (0.2) | 85.8 (0.2) | 85.4 (0.6) | 67.3 (1.1) | 29.4 (0.4) |
| SAT | 17.5 (0.2) | 86.0 (0.0) | **95.1 (0.0)** | 88.9 (0.1) | 87.0 (0.1) | 85.6 (0.1) | 87.2 (0.4) | **70.0 (0.3)** | **34.4 (0.4)** |
| DCM (ours) | **13.9 (0.7)** | **86.5 (0.2)** | 95.0 (0.1) | **89.5 (0.3)** | **88.0 (0.4)** | 86.6 (0.4) | **88.2 (0.8)** | 66.5 (0.3) | 29.8 (1.1) |

Table 16: Selective classification on the Waterbirds spurious correlation dataset. Bold numbers represent superior results, and parentheses show the standard error over 3 random seeds. DCM consistently outperforms all 7 prior methods when the validation and test sets are from different distributions, suggesting that DCM is effective in spurious correlation settings.

| Method | ECE (↓) | Acc (↑) | AUC (↑) | Acc@90 (↑) | Acc@95 (↑) | Acc@99 (↑) | Cov@90 (↑) | Cov@95 (↑) | Cov@99 (↑) |
|---|---|---|---|---|---|---|---|---|---|
| Val = Camelyon17 ID Val-1, Test = Camelyon17 ID Val-2 | | | | | | | | | |
| MSP | 16.3 (10.4) | 81.5 (7.8) | 96.9 (2.2) | 92.0 (5.9) | 90.8 (6.4) | 89.5 (6.6) | 87.2 (10.5) | 78.6 (17.5) | 60.5 (25.5) |
| MaxLogit | 16.4 (10.2) | 81.5 (7.8) | 97.0 (2.2) | 92.2 (5.8) | 91.0 (6.4) | 89.8 (6.5) | 87.8 (10.0) | 79.1 (17.1) | 60.1 (27.0) |
| Ensemble | 15.7 (11.2) | 94.8 (6.4) | 99.1 (2.7) | 96.8 (5.9) | 95.9 (6.7) | 95.1 (6.8) | 100.0 (10.5) | 99.4 (18.1) | 73.2 (27.7) |
| Binary Classifier | 16.3 (9.2) | 89.4 (6.5) | 97.0 (4.5) | 92.3 (5.9) | 91.4 (6.0) | 90.3 (6.7) | 88.1 (10.2) | 79.3 (16.8) | 61.0 (24.2) |
| Fine-Tuning | 2.8 (0.2) | 98.3 (0.2) | **99.8 (0.0)** | **99.7 (0.0)** | 99.4 (0.1) | **98.6 (0.2)** | 100.0 (0.0) | 100.0 (0.0) | **97.3 (0.7)** |
| DG | 3.4 (2.1) | 97.5 (0.4) | **99.8 (0.0)** | 99.6 (0.1) | 99.2 (0.3) | 98.2 (0.4) | 100.0 (0.0) | 100.0 (0.0) | 96.1 (1.7) |
| SAT | **1.6 (0.0)** | 97.7 (0.0) | **99.8 (0.0)** | **99.7 (0.0)** | 99.4 (0.0) | 98.5 (0.0) | 100.0 (0.0) | 100.0 (0.0) | 96.7 (0.3) |
| DCM (ours) | 9.9 (0.3) | 95.3 (0.2) | 99.5 (0.1) | 98.6 (0.2) | 98.0 (0.0) | 96.6 (0.2) | 100.0 (0.0) | 100.0 (0.0) | 82.4 (5.3) |
| Val = Camelyon17 ID Val-1, Test = Camelyon17 ID Val-2 + Camelyon17 OOD Test | | | | | | | | | |
| MSP | 25.9 (6.4) | 66.2 (5.1) | 85.8 (3.7) | 74.1 (5.1) | 73.1 (4.9) | 72.2 (4.8) | 40.6 (7.8) | 29.4 (8.1) | 7.4 (3.3) |
| MaxLogit | 25.9 (6.4) | 66.2 (5.1) | 85.8 (3.7) | 74.2 (5.1) | 73.1 (5.0) | 72.2 (4.8) | 40.7 (7.9) | 29.4 (8.2) | 7.7 (3.6) |
| Ensemble | 19.6 (6.7) | 75.6 (4.6) | 86.5 (4.1) | 78.1 (4.8) | 76.8 (5.2) | 75.8 (4.2) | 25.8 (8.1) | 18.7 (8.4) | 11.5 (3.5) |
| Binary Classifier | 26.3 (6.0) | 72.0 (4.7) | 86.2 (3.3) | 74.4 (5.0) | 73.4 (4.9) | 72.7 (4.4) | 41.0 (8.1) | 29.8 (8.2) | 7.5 (3.5) |
| Fine-Tuning | 20.5 (1.9) | 76.7 (3.4) | 88.9 (2.2) | 79.8 (3.5) | 78.6 (3.4) | 77.6 (3.3) | 44.2 (5.8) | 33.1 (2.8) | 9.7 (6.3) |
| DG | 27.5 (7.3) | 74.0 (5.8) | 88.1 (4.1) | 77.2 (6.5) | 75.8 (6.3) | 74.8 (6.0) | 51.3 (14.3) | 36.4 (9.3) | 6.2 (3.8) |
| SAT | 24.3 (2.1) | 72.1 (1.1) | 86.3 (0.4) | 74.8 (1.1) | 73.8 (1.1) | 73.0 (1.2) | 35.2 (1.3) | 28.3 (1.6) | 15.2 (1.4) |
| DCM (ours) | **8.9 (1.5)** | **80.6 (1.0)** | **93.5 (0.6)** | **85.5 (1.0)** | **83.8 (1.0)** | **82.5 (0.9)** | **74.1 (4.3)** | **50.3 (6.5)** | **16.4 (0.6)** |
| Val = Camelyon17 ID Val-1, Test = Camelyon17 OOD Test | | | | | | | | | |
| MSP | 28.2 (5.4) | 63.1 (4.8) | 82.2 (3.9) | 70.4 (4.8) | 69.5 (4.6) | 68.8 (4.4) | 31.5 (6.8) | 21.8 (7.8) | 2.4 (0.4) |
| MaxLogit | 28.2 (5.4) | 63.1 (4.8) | 82.1 (3.9) | 70.4 (4.8) | 69.5 (4.6) | 68.8 (4.4) | 31.4 (6.9) | 21.9 (7.8) | 2.4 (0.4) |
| Ensemble | 21.1 (5.6) | 71.8 (4.8) | 81.4 (4.4) | 74.0 (5.2) | 72.8 (4.3) | 72.0 (4.7) | 13.4 (7.3) | 9.8 (8.1) | 4.6 (0.3) |
| Binary Classifier | 28.0 (5.3) | 69.0 (5.2) | 82.4 (3.9) | 70.5 (4.4) | 70.1 (4.2) | 69.5 (6.5) | 32.1 (5.6) | 22.1 (7.3) | 2.5 (0.4) |
| Fine-Tuning | 23.6 (2.0) | 72.8 (4.2) | 84.2 (3.8) | 75.4 (4.2) | 74.3 (4.1) | 73.5 (4.0) | 31.4 (6.0) | 21.6 (6.5) | 3.8 (3.1) |
| DG | 31.6 (8.5) | 69.4 (7.5) | 84.8 (5.2) | 72.1 (7.9) | 70.9 (7.7) | 70.1 (7.3) | 43.3 (11.2) | 31.9 (8.0) | 5.8 (4.0) |
| SAT | 21.7 (2.3) | 70.2 (0.7) | 80.3 (0.6) | 71.9 (0.8) | 71.3 (1.0) | 70.8 (1.0) | 20.3 (3.6) | 5.9 (4.0 ) | 0.0 (0.0) |
| DCM (ours) | **11.5 (2.1)** | **78.7 (1.2)** | **91.6 (1.1)** | **82.5 (1.2)** | **80.9 (1.1)** | **79.7 (1.1)** | **62.5 (8.2)** | **40.3 (7.5)** | **9.7 (2.9)** |
| Val = FMoW ID Val, Test = FMoW ID Test | | | | | | | | | |
| MSP | 1.8 (0.4) | 58.4 (1.5) | 81.3 (0.4) | 62.6 (0.1) | 60.9 (0.1) | 58.7 (0.1) | 36.7 (1.3) | 18.4 (5.8) | 3.7 (1.8) |
| MaxLogit | 1.8 (0.4) | 58.4 (0.1) | 80.1 (0.2) | 62.7 (0.2) | 60.6 (0.1) | 58.8 (0.1) | 29.7 (0.8) | 10.7 (2.2) | 0.8 (0.4) |
| Ensemble | **0.8 (0.0)** | **62.5 (0.1)** | **85.5 (0.0)** | **68.4 (0.1)** | **66.1 (0.1)** | **64.2 (0.1)** | **44.8 (0.3)** | **31.5 (0.1)** | **10.6 (0.8)** |
| Binary Classifier | 1.9 (0.4) | 58.4 (0.2) | 82.3 (0.3) | 64.3 (0.1) | 62.0 (0.2) | 60.2 (0.1) | 37.6 (0.5) | 20.9 (4.3) | 3.7 (2.7) |
| Fine-Tuning | 1.2 (0.5) | 59.3 (2.7) | 82.8 (0.9) | 64.0 (1.2) | 61.7 (1.3) | 59.9 (1.2) | 39.5 (2.4) | 27.0 (2.8) | 5.9 (1.2) |
| DG | 1.8 (0.1) | 58.5 (0.4) | 75.8 (0.2) | 62.4 (0.9) | 60.9 (0.5) | 59.9 (0.4) | 12.9 (0.6) | 2.6 (1.8) | 0.1 (0.0) |
| SAT | 1.1 (0.0) | 58.3 (0.5) | 81.1 (0.3) | 63.0 (0.5) | 60.8 (0.5) | 59.1 (0.5) | 33.5 (0.5) | 18.8 (1.0) | 4.3 (0.9) |
| DCM (ours) | 1.1 (0.5) | 59.3 (1.2) | 82.9 (1.1) | 64.2 (1.2) | 61.7 (1.3) | 59.9 (1.1) | 39.4 (2.5) | 26.7 (3.5) | 6.3 (2.0) |
| Val = FMoW ID Val, Test = FMoW ID Test + FMoW OOD Test | | | | | | | | | |
| MSP | 2.3 (0.4) | 51.5 (0.1) | 77.1 (0.5) | 57.9 (0.1) | 55.9 (0.2) | 54.2 (0.0) | 25.4 (2.3) | 11.0 (4.6) | 1.2 (0.6) |
| MaxLogit | 2.3 (0.5) | 51.5 (0.1) | 75.8 (0.1) | 57.8 (0.1) | 55.7 (0.1) | 54.2 (0.0) | 19.4 (0.3) | 4.3 (0.9) | 0.3 (0.2) |
| Ensemble | **1.3 (0.0)** | **56.5 (0.0)** | **81.7 (0.0)** | **63.2 (0.0)** | **61.2 (0.0)** | **59.4 (0.0)** | **35.6 (0.2)** | **24.3 (0.1)** | **5.6 (0.2)** |
| Binary Classifier | 2.5 (0.4) | 53.8 (0.1) | 78.0 (0.4) | 59.3 (0.0) | 57.3 (0.0) | 55.6 (0.0) | 27.5 (1.5) | 9.3 (6.2) | 1.2 (0.9) |
| Fine-Tuning | 1.7 (0.3) | 54.2 (2.3) | 78.6 (0.8) | 58.6 (1.2) | 56.5 (1.1) | 54.8 (1.1) | 30.8 (2.1) | 19.1 (1.7) | 3.0 (0.3) |
| DG | 2.2 (0.0) | 54.0 (0.3) | 71.6 (0.2) | 57.5 (0.3) | 56.1 (0.2) | 55.1 (0.2) | 5.0 (0.2) | 0.2 (0.1) | 0.0 (0.0) |
| SAT | 1.4 (0.0) | 53.7 (0.4) | 76.7 (0.2) | 57.8 (0.4) | 55.8 (0.4) | 54.3 (0.4) | 24.8 (0.3) | 11.2 (0.8) | 0.5 (0.2) |
| DCM (ours) | 1.5 (0.3) | 54.6 (1.7) | 78.9 (1.1) | 58.8 (1.3) | 56.7 (1.3) | 55.0 (1.3) | 30.7 (2.0) | 20.1 (2.2) | 3.8 (1.1) |
| Val = FMoW ID Val, Test = FMoW OOD Test | | | | | | | | | |
| MSP | 2.6 (0.5) | 50.9 (2.7) | 74.5 (0.6) | 55.2 (0.2) | 53.4 (0.3) | 52.0 (0.1) | 20.6 (3.2) | 8.4 (3.5) | 1.3 (0.4) |
| MaxLogit | 2.6 (0.5) | 50.7 (0.1) | 73.3 (0.2) | 55.2 (0.0) | 53.3 (0.1) | 51.9 (0.1) | 13.7 (0.4) | 3.2 (0.8) | 0.3 (0.1) |
| Ensemble | **1.6 (0.0)** | **55.0 (0.1)** | **79.5 (0.0)** | **60.7 (0.1)** | **58.6 (0.1)** | **57.0 (0.1)** | **31.1 (0.1)** | **20.8 (0.4)** | **3.3 (0.6)** |
| Binary Classifier | 2.8 (0.5) | 51.7 (0.0) | 75.6 (0.5) | 56.8 (0.1) | 54.9 (0.0) | 53.3 (0.0) | 21.1 (3.8) | 7.6 (4.8) | 1.0 (0.4) |
| Fine-Tuning | 2.0 (0.3) | 51.8 (1.1) | 76.2 (0.8) | 56.0 (0.9) | 53.9 (0.9) | 52.4 (1.0) | 26.8 (1.4) | 14.8 (1.3) | 1.7 (0.1) |
| DG | 2.5 (0.0) | 51.9 (0.1) | 69.2 (0.3) | 54.9 (0.2) | 53.7 (0.1) | 52.6 (0.0) | 2.7 (0.8) | 0.1 (0.1) | 0.0 (0.0) |
| SAT | **1.6 (0.0)** | 51.0 (0.4) | 74.1 (0.2) | 55.1 (0.4) | 53.2 (0.3) | 51.8 (0.3) | 20.1 (0.4) | 7.1 (1.1) | 0.3 (0.1) |
| DCM (ours) | 1.8 (0.3) | 51.9 (1.7) | 76.4 (1.1) | 56.2 (1.4) | 54.1 (1.3) | 52.4 (1.2) | 26.8 (1.6) | 16.3 (1.7) | 2.4 (0.5) |

Table 17: Selective classification on the Camelyon17 and FMoW domain shift datasets. Bold numbers represent best performance, and parentheses show the standard error over 3 random seeds. On Camelyon17, DCM consistently outperforms all 7 prior methods when the validation and test sets are from different distributions. On FMoW, DCM has the second-highest AUC after Ensemble, while using only 1/5 of the compute.

