# OpenReview forum: "Conservative Prediction via Data-Driven Confidence Minimization"
_TMLR — Accepted by TMLR_

### Review · Reviewer_jSVF · 2024-02-07

**Summary Of Contributions:**

- The authors present a training-based framework to improving uncertainty estimation termed DCM, where the model is trained to minimise its confidence on an "uncertainty dataset" which should contain samples that are representative of "unknowns" encountered at test time.

- They provide some theoretical results regarding DCM of limited practical utility.

- They show promising results for their approach in the scenarios of selective classification under distribution shift, and very strong results for OOD detection.

**Audience:**

Yes

**Claims And Evidence:**

No

**Requested Changes:**

**Critical** (I will recommend acceptance if the below changes/additions are made)

1. Top 1 accuracy results should be visible in all tables in the main paper.

2. DCM should be better compared with [1]

	- DCM for OOD detection should be clarified as an existing baseline approach in [1]

	- How top 1 accuracy differs to WOODS

	- How DCM outperforms WOODS in OOD detection here but not in [1]

3. Short discussion on rejecting *both* ID misclassifications *and* OOD samples with accompanying references [2,3] and how this problem setting may relate to DCM (additional experiments not necessary, sufficient to leave as possible future work).

4. Clarification on the limitations of Proposition 4.2 with regards to image-space distances.

**Non-Critical**

1. Notation that is clearer and less confusing to the reader.

2. More useful insight as to why DCM improves SC (and top 1 accuracy) on covariate-shifted data.

3. Ideally I would prefer Section 4 to be rewritten to instead illustrate the intuition behind DCM rather than contain somewhat irrelevant theory.

**Strengths And Weaknesses:**

**Strengths**

- The authors provide a wide suite of experiments on both small-scale (CIFAR) and higher-resolution (ImageNet-1k) data.

- Ample details with regards to the aforementioned experiments are provided to aid reproducibility.

- DCM performs exceptionally well on the task of OOD detection, demonstrating

	- the advantage of using OOD data from the test-time distribution for confidence minimisation

	- that it is not necessary to exclude ID data from confidence minimisation

- DCM shows promising performance for selective classification under covariate shift

**Weaknesses**

1. There is a lack of transparency with regards to the effect of DCM on in-distribution top 1 accuracy. This is relegated to Appendix J when it should be clearly available to the reader. That is to say a ~-3% decrease for CIFAR-100 and ImageNet-1k.

2. Positioning with regards to Katz-Samuels et al [1].

    - The authors treat WOODS from [1] as a competing method but disregard the fact that in [1] DCM (both energy and softmax) is the baseline approach as the uncertainty dataset in [1] is constructed in the same way as this paper. In [1] WOODS is shown to outperform DCM, so there should be discussion in this paper as to why the reverse occurs here.

    - One of the advantages of WOODS is the explicit objective of preserving ID top 1 accuracy. This is not discussed in the paper and is especially relevant as DCM appears to reduce top 1 accuracy (see 1.)

3. My impression is that the theoretical insights don't apply to real-world image data (the notation is quite confusing here so I welcome the authors to correct me if I'm wrong).

	- Proposition A.4 in Appendix A.3 applies to $\delta$-balls around the data points in the uncertainty set.

	- My understanding is that it is suggested test-time ID and OOD images will fall into $\delta$-balls near the ID and OOD images in the uncertainty set.

	- Distances in the image space are somewhat meaningless semantically (adversarial attacks rely on this). For example, translation can move an image a large distance whilst barely changing the semantics. Drawing thin legs on a snake makes it a lizard but moves the image a small distance.

	- Thus it doesn't make sense to assume that any test-time ID/OOD samples will fall within $\delta$-balls near the corresponding uncertainty set images.

	- I still think that the intuition presented however is perfectly valid in my eyes, i.e. cross entropy cancels out the effect of confidence minimisation on the ID samples, allowing ID and OOD to be separated.

4. There is no acknowledgement of the problem scenario where *both* OOD and misclassified samples should be detected [2,3]. Although it is not necessary to fully explore it in this paper, it should be at the least discussed as

	- There is potential to expand the approaches in this paper to the setting above.

	- DCM solely for OOD or SC may have an adverse effect on the other task. This is salient as one way to improve OOD detection is to (relatively) increase confidence on ID misclassifications [3].

5. Although DCM performs well for selective classification under covariate shift, there is little discussion as to why this might be the case. The authors do not provide any useful insight or disentangle the contributions of better top1 accuracy and better misclassification detection as both can improve SC.

6. Confusing/imprecise notation/terminology

	- "OOD" used to refer to both semantic and covariate shift [4] without distinction.

	- $D_u, D_{unc}, D_{tr}, D^{train}, D^{test},  D^{test}_{unk}$ (which is used for the *training* uncertainty set?)

    - "Calibrate" in 3.1 can be easily misunderstood as DCM is orthogonal to classifier confidence calibration as defined in Section 6.

**Queries**

1. The results for the Ensemble on CIFAR-10 seem to be off (no accuracy change vs single model, Tab. 17). I would encourage the authors to double-check their implementation as in my experience ensembling should give ~+1% accuracy boost on that dataset.



[1] Katz-Samuels et al. Training OOD Detectors in their Natural Habitats, ICML 2022

[2] Jaeger et al. A Call to Reflect on Evaluation Practices for Failure Detection in Image Classification, ICLR 2023

[3] Xia and Bouganis. Augmenting Softmax Information for Selective Classification with Out-of-Distribution Data, ACCV 2022

[4] Yang et al. Generalized Out-of-Distribution Detection: A Survey, 2021

---

> ### Author Response · Authors · 2024-04-17
> **Response to Reviewer jSVF**
>
> Thank you for your detailed and insightful feedback on our work! We respond to your comments below:
>
> > Top 1 accuracy results should be visible in all tables in the main paper.
>
> Thank you for pointing this out. We have added top-1 accuracy results to all tables in the main paper. While DCM does trade off a small amount of ID accuracy for improved uncertainty estimation, we see that it actually improves even top-1 accuracy in distribution shift conditions. Please note that we omitted ID test accuracies for the Binary Classifier and ERD baselines in OOD detection, as these are trained with objectives for which classification accuracy is not meaningful.
>
> > DCM should be better compared with [1]. DCM for OOD detection should be clarified as an existing baseline approach in [1]; How top 1 accuracy differs to WOODS; How DCM outperforms WOODS in OOD detection here but not in [1]
>
> There are several differences between our experimental setup and the experimental setup in WOODS. We closely follow the setup established by Hendrycks et al [2].
>
> - **Use of Image Augmentations**: Unlike DCM, the baseline approach in [1] does not use random augmentations on the unlabeled set to prevent overfitting (https://github.com/jkatzsam/woods_ood/blob/80d706a3739b14498ac99647e42855ad14e68562/CIFAR/make_datasets.py#L59). Thus DCM is not quite the existing baseline approach in [1].
> - **Different Sampling Ratios**: At each gradient update, WOODS computes the combined objective using a 10:1 ratio of ID:uncertainty set, whereas we use a 1:1 sampling ratio.
> - **Different Validation Sets**: WOODS uses a mixed ID-OOD validation set for hyperparameter tuning and early stopping. DCM and Outlier Exposure only use an ID validation set.
>
> The performance differences between our approach and WOODS can be attributed to these factors.
>
> > Short discussion on rejecting both ID misclassifications and OOD samples with accompanying references [2,3] and how this problem setting may relate to DCM (additional experiments not necessary, sufficient to leave as possible future work).
>
> Thank you for the suggestion. In our edited submission, we highlight that this is a natural extension of the DCM framework to the Related Work and Conclusion (Sections 5 and 7).
>
> > Clarification on the limitations of Proposition 4.2 with regards to image-space distances.
>
> We broadly agree with the reviewer’s assessment; the main purpose of Proposition 4.2 is to formalize the intuitive argument that the cross-entropy cancels out confidence minimization near ID regions. However, we also note that when fine-tuning the last layers of a pre-trained network, Proposition 4.2 directly applies to feature-space distances, which are known to reflect semantic relations. We have edited Section 4 to clarify these points.
>
> > More useful insight as to why DCM improves SC (and top 1 accuracy) on covariate-shifted data.
>
> Thank you for pushing us to provide a more intuitive explanation for DCM’s OOD performance gains. We think this is because encouraging the model to be uncertain on challenging ID examples leads to representations that are less prone to being overconfident on unusual examples while focusing on reliable predictive signals. This aligns with recent work showing the benefits of learning invariant representations for OOD generalization.
>
> > The results for the Ensemble on CIFAR-10 seem to be off (no accuracy change vs single model, Tab. 17). I would encourage the authors to double-check their implementation as in my experience ensembling should give ~+1% accuracy boost on that dataset.
>
> Thank you for your observation. We corrected an error in our Ensemble experiments for CIFAR-10 in Table 17, where we had incorrectly ensembled the same pretrained model instead of different ones. This adjustment led to a ~1% increase in accuracy for the Ensemble on CIFAR-10. We’ve updated the table and relevant sections of our manuscript to reflect this change.

---

> > ### Comment · Reviewer_jSVF · 2024-04-23
> >
> > Thanks for taking the time to update the manuscript and clarify my queries. I'm happy with the changes that have been made. In particular, I agree that reasoning about feature-space distances greatly strengthens the relevance of Proposition 4.2. I just have a couple of small additional requests:
> >
> > - Add your clarification highlighting the differences with regards to WOODS to the Appendix.
> > - Add a corresponding reference after "feature-space distances, which are known to reflect semantic relations".
> >
> > I'd also like to know why the accuracy results in Table 3 (79.6) do not match with Table 12 (81.42).

---

> > > ### Author Response · Authors · 2024-04-24
> > > **Response to Reviewer jSVF**
> > >
> > > > Add your clarification highlighting the differences with regards to WOODS to the Appendix.
> > > > Add a corresponding reference after "feature-space distances, which are known to reflect semantic relations".
> > >
> > > Thank you for these suggestions. We’ve added the WOODS clarification to Appendix D.1 and cited [1,2] in Section 4.
> > >
> > > > I'd also like to know why the accuracy results in Table 3 (79.6) do not match with Table 12 (81.42).
> > >
> > > Table 3 presents numbers from the NPOS paper (Table 2 of NPOS: https://arxiv.org/abs/2303.02966). The base model weights from NPOS were not available, so Table 12 reports results using an alternative open-source model with the same architecture, pretrained on ImageNet (https://pytorch.org/vision/main/models/generated/torchvision.models.vit_b_16.html), and evaluated on 10,000 class-balanced ImageNet test images.
> > > We’ve removed Tables 11 and 12 from the appendix as ID accuracy is now adequately covered in Tables 2, 3, and 4.
> > >
> > > References
> > >
> > > [1] Upchurch, Paul, Jacob Gardner, Geoff Pleiss, Robert Pless, Noah Snavely, Kavita Bala, and Kilian Weinberger. "Deep feature interpolation for image content changes." In Proceedings of the IEEE conference on computer vision and pattern recognition, pp. 7064-7073. 2017.
> > >
> > > [2] Wang, Yulin, Xuran Pan, Shiji Song, Hong Zhang, Gao Huang, and Cheng Wu. "Implicit semantic data augmentation for deep networks." Advances in Neural Information Processing Systems 32 (2019).

---

> ### Comment · Reviewer_jSVF · 2024-04-24
>
> Thanks for the additional changes and clarification. I am now happy with the state of the manuscript. Hopefully the authors feel that their work has been improved through the review process as well.

---

> ### Author Response · Authors · 2024-04-24
> **Response to Reviewer jSVF**
>
> We are happy that we have addressed all of the reviewer's concerns! We also thank reviewer jSVF for their thorough and thoughtful review, it has helped us improve our work.

---

### Review · Reviewer_GpJw · 2024-02-17

**Summary Of Contributions:**

The paper introduces Data-Driven Confidence Minimization (DCM) as a unified method for conservative prediction in selective classification and out-of-distribution (OOD) detection. It outlines a practical approach to construct an uncertainty dataset tailored to each problem setting. For selective classification (SC), the uncertainty dataset comprises misclassified examples from a separate validation set, mirroring potential misclassifications during testing. In OOD detection, the uncertainty dataset consists of unlabeled examples from the test distribution, including both known and unknown instances. The authors acknowledge that unlabeled test examples might not always be accessible but highlight real-world scenarios, such as unannotated medical data from new hospitals, where they are present. The theoretical guarantees suggest that minimizing the DCM objective can provably separate known and unknown data with an appropriate threshold on the maximum softmax probability. The authors evaluate the DCM method empirically for SC and OOD detection. In SC, it achieves a 2.3% increase in performance under distribution shift conditions and surpasses an ensemble of models despite lower computational costs. In OOD detection, DCM outperforms Outlier Exposure by reducing False Positive Rates by 6.3% and 58.1% on CIFAR-10 and CIFAR-100 datasets, respectively. Moreover, it demonstrates strong performance in challenging near-OOD detection settings, achieving higher AUROC values than state-of-the-art methods.

**Audience:**

Yes

**Broader Impact Concerns:**

N/A.

**Claims And Evidence:**

Yes

**Requested Changes:**

1. Please state the assumptions clearly in the main paper.
2. If possible, unify Algorithm 1 and 2 in a single procedure that can give a single model that would work for both OOD and SC.
3. There are works providing threshold estimation procedures [1,2] for selective classification. I suppose these methods could be utilized in this work and might improve the performance further.


[1] https://openreview.net/pdf?id=RUCFAKNDb2

[2] https://arxiv.org/pdf/1705.08500.pdf

**Strengths And Weaknesses:**

Strengths:

1.  The introduction of Data-Driven Confidence Minimization (DCM) provides a unified method for selective classification and out-of-distribution (OOD) detection using held-out validation and samples from OOD, offering a fresh perspective on these tasks.

2. The authors provide theoretical guarantees and extensive empirical evaluation supporting the efficacy of the DCM objective in separating known and unknown data.

Weaknesses:

1. There are two separate versions of DCM each solving a different objective for selective classification(SC)  and OOD detection. In practice, we would like to train a classifier once and ensure that during deployment it is doing both SC and OOD detection.

2. It is not clear how is the confidence threshold selected for selective classification and what kind of error and coverage guarantees can be provided for this setting when using DCM.

3. The method requires validation and unknown samples but it is not clear how many such samples are required by the method to ensure certain performance guarantees.

---

> ### Author Response · Authors · 2024-04-17
> **Response to Reviewer GpJw**
>
> Thank you for your detailed and insightful feedback on our work! We respond to your comments below:
>
> > There are two separate versions of DCM each solving a different objective for selective classification(SC) and OOD detection. In practice, we would like to train a classifier once and ensure that during deployment it is doing both SC and OOD detection.
>
> We’ve added a brief discussion on rejecting both ID misclassifications and OOD samples with accompanying references [2,3] to Section 5 (Related Work). We also discuss how DCM can be extended to this problem setting as future work in Section 7 (Conclusion).
>
> > If possible, unify Algorithm 1 and 2 in a single procedure that can give a single model that would work for both OOD and SC.
>
> You raise an important point about the desirability of a single unified model for both notions of model conservativeness. In the paper, we treat these as distinct settings and offer two variants of the same workflow for each in order to focus on the specific metrics relevant to each problem. Algorithm 2 is specifically optimized for SC metrics, directly addressing the nuances of making accurate and confident predictions on known samples. We maintain separate algorithms to ensure optimal performance for each task.
>
> > It is not clear how is the confidence threshold selected for selective classification and what kind of error and coverage guarantees can be provided for this setting when using DCM.
> > There are works providing threshold estimation procedures [1,2] for selective classification. I suppose these methods could be utilized in this work and might improve the performance further.
>
> The confidence threshold is chosen to achieve a desired accuracy or coverage based on the specific application's risk tolerance. This can be done using a held-out validation set. While our paper does not focus on explicit coverage guarantees, we show that the confidence from a model that minimizes our loss will maximally separate ID from OOD examples. We believe that providing theoretical guarantees is somewhat orthogonal to the goal of our work, which aims to develop a method that enables conservative prediction in real datasets. We believe extending our work while incorporating methods with coverage guarantees is an exciting direction for future work.
>
> > The method requires validation and unknown samples but it is not clear how many such samples are required by the method to ensure certain performance guarantees.
>
> In our experiments, we find that using an uncertainty set that is around 10% of the training set size is sufficient.
>
> > Please state the assumptions clearly in the main paper.
>
> Thank you for the feedback. We have revised our paper to more clearly state these assumptions in Section 2.1 and Section 3.
>
>
> References
>
> [1] Angelopoulos, Anastasios N., and Stephen Bates. "A gentle introduction to conformal prediction and distribution-free uncertainty quantification." arXiv preprint arXiv:2107.07511 (2021).

---

> > ### Author Response · Authors · 2024-05-01
> >
> > Dear Reviewer GpJw,
> >
> > Thanks a lot for your insightful review, it helped us improve our work! We wanted to follow-up with you to see if our rebuttal answers all your questions: we are happy to engage in a discussion and answer any other questions you might have about our work!
> >
> > Thanks!

---

### Review · Reviewer_6SyM · 2024-03-25

**Summary Of Contributions:**

This work proposes Data-Driven Confidence Minimization (DCM) frameworks that minimizes confidence on an uncertainty dataset curated during training to help make better predictions on unknown inputs at test time. It applies the DCM frameworks to Out-of-Distribution (OOD) detection and selective classification problems. At the heart of this framework lies two components: (a) curate an uncertainty set on which the model makes uncertain predictions during training, and (b) minimize a training object that balances target objective (such as classification error on train set) and confidence on the uncertainty set. Such a minimizer aims to make less confidence predictions on inputs where it is uncertain, and make confident predictions on the inputs where it is certain.

DCM works by first pre-training the model to achieve a good performance on the training set (albeit at the cost of overfitting the training set). Then, it proceeds to curate an uncertainty set, which is used during the fine-tuning phase, where DCM minimizes the target objective along with minimizing the confidence on the uncertainty set.

This work performs empirical evaluations against a variety of Selective Classification and OOD detection datasets to test the proposed DCM framework.

**Audience:**

Yes

**Claims And Evidence:**

Yes

**Requested Changes:**

Questions for Authors:
------------
- What stops the Algorithm 1 to simply overfit the validation examples?
- In practice, how do you justify the assumption that the uncertainty set consists of unknown examples?
- Any reasons to not include CIFAR-100 for Selective Classification evaluation?
- What is the inference procedure? How does the method reject prediction during the test time?
- Do you have any practical advice for curating the uncertainty dataset on a new problem setting? How large should this dataset be compared to the training set?
- In Table 1, what inference one should make from the ID part of the table, where many baselines are significantly better than proposed DCM framework?
- In Table 1, why is the ensemble method consistently amongst the top performing schemes?
- What is the computational overhead associated with the DCM framework? How much is the additional training time compared to the baselines?

**Strengths And Weaknesses:**

Strengths:
-----------
- Proposed DCM framework seems promising in that it combines the target objective along with the minimization of the confidence on uncertainty dataset, enabling lower confidence on uncertain examples.

- Empirical evaluation shows that DCM outperforms existing approaches in OOD detection.

Weaknesses:
-----------
- At the heart of the DCM framework is choice of the uncertainty dataset, the paper lacks discussion on the choice of this dataset in practice. How big should this dataset be? How to curate this dataset on a new problem? Since a large capacity neural network can overfit even an uncertainty set without some considerations.

- Some inconsistencies in the empirical evaluation (see below)

---

> ### Author Response · Authors · 2024-04-17
> **Response to Reviewer 6SyM**
>
> Thank you for your detailed and insightful feedback on our work! We respond to your comments below:
>
> > What stops the Algorithm 1 to simply overfit the validation examples?
>
> This is an important question. While a high-capacity model could theoretically overfit the uncertainty set, standard methods of preventing overfitting are effective here. Specifically, we perform early stopping with respect to ID loss and use random image augmentations on the uncertainty dataset. We also note that we are jointly optimizing the main cross-entropy objective with the confidence objective on the uncertainty set. We will clarify this in the text.
>
> > In practice, how do you justify the assumption that the uncertainty set consists of unknown examples?
>
> We note that the uncertainty set need not consist entirely of unknown examples; this is a central point in our paper. The uncertainty set only needs to contain some unknown examples. Such sets are often readily available in real-world settings. We also show that DCM is effective in transductive settings (Tables 5 & 6). Given a batch of unlabeled test examples, one can fine-tune with DCM on the unlabeled test set itself and then run inference on the same test set to determine which samples are OOD.
>
> > What is the inference procedure? How does the method reject prediction during the test time?
>
> At inference time, we compute the confidence of the model’s prediction for each input and reject all inputs below some confidence threshold. This threshold is a hyperparameter that controls the coverage-risk tradeoff and can be set based on the risk tolerance of the application.
>
> > Do you have any practical advice for curating the uncertainty dataset on a new problem setting? How large should this dataset be compared to the training set?
>
> Our analysis and experiments suggest the following guidelines for curating the uncertainty set:
> For OOD detection: the uncertainty set should contain some OOD examples. It can include ID examples without compromising performance.
> For selective classification: hold out a small validation set during initial training and use the misclassified examples as the uncertainty set.
> In both settings, we find that using an uncertainty set that is around 10% of the training set size is sufficient.
>
> > In Table 1, what inference one should make from the ID part of the table, where many baselines are significantly better than proposed DCM framework?
>
> Many baselines (e.g., fine-tuning on the ID validation set) overfit to the ID distribution. By minimizing confidence on misclassified ID examples, DCM is less prone to overfitting ID.
>
> > In Table 1, why is the ensemble method consistently amongst the top performing schemes?
>
> Ensemble models have been shown to outperform single models on OOD inputs because they cancel out the errors of individual models. As noted in prior works, such aggregation reduces overfitting and improves OOD generalization [1].
>
> > Any reasons to not include CIFAR-100 for Selective Classification evaluation?
>
> We chose to focus on expanding our analysis to include subpopulation and real-world domain shifts, using Waterbirds, WILDS-Camelyon17, and WILDS-FMoW. Prior works on selective classification [2] also did not evaluate on CIFAR-100.
>
> > What is the computational overhead associated with the DCM framework? How much is the additional training time compared to the baselines?
>
> In our experiments, the DCM framework increased training time by up to 10% compared to comparable baselines. This modest overhead stems from the additional network passes needed to minimize confidence on the uncertainty set.
>
> References
>
> [1] Lakshminarayanan, Balaji, Alexander Pritzel, and Charles Blundell. "Simple and scalable predictive uncertainty estimation using deep ensembles." Advances in neural information processing systems 30 (2017).
>
> [2] Liu, Ziyin, Zhikang Wang, Paul Pu Liang, Russ R. Salakhutdinov, Louis-Philippe Morency, and Masahito Ueda. "Deep gamblers: Learning to abstain with portfolio theory." Advances in Neural Information Processing Systems 32 (2019).

---

> > ### Comment · Reviewer_6SyM · 2024-05-05
> > **Rebuttal Acknowledgement**
> >
> > Thanks for the detailed rebuttal and providing clarifications to the operating procedure of the proposed scheme.

---

> > > ### Author Response · Authors · 2024-05-06
> > >
> > > We thank reviewer 6SyM for their thoughtful suggestions that helped us improve our paper! We also are happy that we have addressed the reviewer's concerns.

---

> ### Author Response · Authors · 2024-05-01
>
> Dear Reviewer 6SyM,
>
> Thanks a lot for your insightful review, it helped us improve our work! We wanted to follow-up with you to see if our rebuttal answers all your questions: we are happy to engage in a discussion and answer any other questions you might have about our work!
>
> Thanks!

---

### Author Response · Authors · 2024-04-17
**General response and new changes**

We thank the reviewers for their detailed and insightful comments on our work. We summarize the main paper revision here, and respond to individual reviewers below. All revisions are in blue text.

- **Top-1 Accuracies.** We’ve added top-1 accuracies to all tables in the main paper.

- **Clarification on confidence thresholding.** During inference, we compute each input's model confidence and reject those below a certain threshold, a hyperparameter set based on the application’s risk tolerance. This threshold is a hyperparameter that controls the coverage-risk tradeoff and can be set based on the risk tolerance of the application. In our experiments, the confidence threshold is chosen to achieve a desired accuracy or coverage based on the specific application's risk tolerance, following prior works on selective classification [1,2,3]. This can be done using a held-out validation set. Integrating coverage guarantees represents a promising area for future research. We have clarified these points in Sections 2.1 and 3 of our revised paper.

- **Rejecting both ID misclassifications and OOD samples.** We’ve added a brief discussion on this problem setting with accompanying references.

- **Clarifications on theory.** We've elaborated on the theoretical assumptions underlying our models, particularly in relation to the utility and limitations of our propositions about image-space distances. Additionally, we've included a more intuitive explanation of how DCM enhances performance under distribution shifts.

References

[1] Geifman, Yonatan, and Ran El-Yaniv. "Selective classification for deep neural networks." Advances in neural information processing systems 30 (2017).

[2] Liu, Ziyin, Zhikang Wang, Paul Pu Liang, Russ R. Salakhutdinov, Louis-Philippe Morency, and Masahito Ueda. "Deep gamblers: Learning to abstain with portfolio theory." Advances in Neural Information Processing Systems 32 (2019).

[3] Huang, Lang, Chao Zhang, and Hongyang Zhang. "Self-adaptive training: beyond empirical risk minimization." Advances in neural information processing systems 33 (2020): 19365-19376.

---

### Decision · Action_Editor_MeNE · 2024-05-05

**Recommendation:** Accept with minor revision

**Comment:**

After review and author feedback, the reviewers think that the presented DCM method seems to be generally useful and could potentially be widely applicable to problems related to uncertainty estimation in OOD settings. Although they also pointed out that the experimental gains in the experiments against other baselines are not very significant.

One reviewer also mentioned that the paper could benefit from a better explanation of the intuition behind the method, especially on the analysis of its experimental performance. I would encourage the authors to make an effort on this in revision.

As a side note, I would recommend a short discussion on data augmentation in uncertainty quantification and OOD detection context. E.g., see this paper https://proceedings.mlr.press/v115/hafner20a.html

**Audience:**

Robust ML researchers, researchers interested in uncertainty quantification, distribution shift, etc.

**Claims And Evidence:**

This work proposes Data-Driven Confidence Minimization (DCM) to improve predictions in OOD settings. The idea is to curate an "uncertainty dataset" where the models are trained to make uncertain predictions on these datapoints. A training objective is also proposed to balances task loss and uncertainty quality on the uncertainty dataset. Experiments compare a variety of selective classification and OOD detection methods to demonstrate the effectiveness of DCM.